# One-stone-for-two-birds strategy to attain beyond 25% perovskite solar cells

Tinghuan Yang[1,4], Lili Gao[1,4], Jing Lu[1], Chuang Ma[1], Yachao Du[1], Peijun Wang[2], Zicheng Ding[1], Shiqiang Wang[1], Peng Xu[2], Dongle Liu[1], Haojin Li[1], Xiaoming Chang[1], Junjie Fang[1], Wenming Tian ✉[2], Yingguo Yang[3], Shengzhong (Frank) Liu ✉[1,2] & Kui Zhao[1] ✉

Even though the perovskite solar cell has been so popular for its skyrocketing power conversion efficiency, its further development is still roadblocked by its overall performance, in particular long-term stability, large-area fabrication and stable module efficiency. In essence, the soft component and ionic–electronic nature of metal halide perovskites usually chaperonage large number of anion vacancy defects that act as recombination centers to decrease both the photovoltaic efficiency and operational stability. Herein, we report a one-stone-for-two-birds strategy in which both anion-fixation and associated undercoordinated-Pb passivation are in situ achieved during crystallization by using a single amidino-based ligand, namely 3-amidinopyridine, for metal-halide perovskite to overcome above challenges. The resultant devices attain a power conversion efficiency as high as 25.3% (certified at 24.8%) with substantially improved stability. Moreover, the device without encapsulation retained 92% of its initial efficiency after 5000 h exposure in ambient and the device with encapsulation retained 95% of its initial efficiency after >500 h working at the maximum power point under continuous light irradiation in ambient. It is expected this one-stone-for-two-birds strategy will benefit large-area fabrication that desires for simplicity.

Organic–inorganic hybrid perovskite solar cells (PSCs) have attracted significant attention from researchers since their first demonstration in 2009[1–3] due to the low-cost solution processing[4], tunable bandgap[5], high absorption coefficient[6], low recombination rate[7], and high mobility of charge carriers[8]. The power conversion efficiency (PCE) of single-junction PSCs has rapidly increased from 3.8% to a certified value of 25.7%[9–11]. The long-term operational stability of unencapsulated PSCs has also exceeded 1000 h in full sunlight after the interface engineering of device structures and molecular passivation of the perovskite layer[2,12–14]. Therefore, PSCs represent a promising next-generation photovoltaic technology.

Over the past few years, it has been demonstrated that the elimination of deep-level defects, which act as detrimental nonradiative recombination centers, is critical for realizing high-performance solar cells[15–18]. To date, iodide anions (I⁻) vacancy defects constitute the majority of non-radiative recombination centers in the FAPbI₃ perovskite layer that are very difficult to mitigate[19,20]. These defects are mainly caused by iodine losses during film fabrication or the I⁻ ion

[1]Key Laboratory of Applied Surface and Colloid Chemistry, Ministry of Education; Shaanxi Key Laboratory for Advanced Energy Devices; Shaanxi Engineering Lab for Advanced Energy Technology; Institute for Advanced Energy Materials; School of Materials Science and Engineering, Shaanxi Normal University, Xi'an 710119, China. [2]Dalian National Laboratory for Clean Energy; State Key Laboratory of Molecular Reaction Dynamics and the Dynamic Research Center for Energy and Environmental Materials; iChEM, Dalian Institute of Chemical Physics, Chinese Academy of Sciences, Dalian 116023, China. [3]Shanghai Synchrotron Radiation Facility (SSRF), Zhangjiang Lab, Shanghai Advanced Research Institute, Chinese Academy of Sciences, Shanghai 201204, China. [4]These authors contributed equally: Tinghuan Yang, Lili Gao. ✉e-mail: tianwm@dicp.ac.cn; liusz@snnu.edu.cn; zhaok@snnu.edu.cn

migration during device operation, which produces deep-level defects and directly leads to the degradation of photoelectric properties[15–18]. Furthermore, the I⁻ ions desorbed from the inorganic framework can be easily oxidized to I⁰ species, which initiate chemical chain reactions accelerating the formation of deep-level defects in perovskite layers[21]. This phenomenon is detrimental for devices operating under complex conditions (including high temperature, continuous light illumination, electrical bias, or their combination) and negatively affects the long-term stability of operational PSCs[22,23]. Therefore, inhibiting the formation and migration of I⁻ vacancy defects is critical for stabilizing photoactive perovskite layers and preserving their good photoelectric properties.

Several attempts have been made to passivate anion vacancy defects through the introduction of additives such as quaternary ammonium halide anions and cations[24], caffeine and theobromine[19], CsI–DB21C7 complex[25], naphthalene-1,8-dicarboximide and perylene-3,4-dicarboximide[26]. However, these additives are typically used to passivate undercoordinated Pb²⁺ ions after anion species have already escaped from the crystal lattice. To inhibit the formation of anion vacancy defects radically, anions should be pinned to the crystal lattice without the possibility of escaping, which can be theoretically realized through a judicious chemical design by molecular engineering. There are five potential benefits of this approach. First, the introduced materials should be strongly bonded to the Pb–I framework to pin anions to the crystal lattice and suppress the formation of anion vacancies in the source. Second, strong chemical bonding should localize anions that have already escaped from the crystal lattice, which would inhibit the migration of delocalized anions. Third, the space occupied by the introduced materials in the Pb–I framework should not be too large to ensure efficient charge transport between different Pb–I frameworks. Fourth, the introduced materials should be stable under thermal stress and illumination. Fifth, the introduced materials should act as growth-controlling agents to promote the crystallization of perovskites. In previous studies, various additives such as phosphonopropionic acid[13] and amino-based organic ligands[27–29] were used as sacrificial agents to inhibit the formation of anion vacancy defects. However, phosphonopropionic molecules tend to degrade upon heating, while amino-based organic ligands cannot form strong chemical bonds to pin anions to the crystal lattice. Furthermore, the large barrier layer of these organic compounds in the Pb–I framework negatively affected charge transport and led to the formation of a charge extraction barrier, making them unsuitable for high-efficiency stable solar cells.

In this work, we propose a rational design strategy for anion fixation and suppressing the formation of anions vacancy defects synergistically by a sustainable pinning effect while retaining the efficient charge transport using amidino-based organic molecules to construct highly efficient and stable PSCs. 3-amidinopyridine (3AP) molecules form strong chemical bonds with the Pb–I framework to effectively pin anions and considerably increase the energy barrier of the formation and migration of anion vacancies. Therefore, this one-stone-for-two-birds strategy enables us to achieve a PCE as high as 25.3% (certified at 24.8%) and significantly improved operation stability following the ISOS-L-1 stability protocol. The present work paves the way for the anion-vacancy defect engineering via molecule–perovskite coordination, providing an effective and simple solution towards efficient and stable perovskite optoelectronics.

## Results
### Molecule–Perovskite coordination
3AP was selected as an additive to pin anions to the crystal lattice, its chemical structure, Nuclear magnetic resonance hydrogen spectroscopy (¹H-NMR) and High-resolution mass spectral (HRMS) are shown in Fig. 1a, Supplementary Figs. 1 and 2, respectively. The amidino-based terminal group and –NH–C group in the pyridine ring provide strong

coordination with Pb–I inorganic frameworks and modulate the Pb–I interlayer distance. By analyzing (3AP)PbI₄ single crystals (Supplementary Fig. 3 and Supplementary Table 1)[30], we found that 3AP molecules were arranged in parallel and anti-symmetrically between Pb–I frameworks with a short interlayer distance of 3.45 Å, which is much shorter than that obtained for previously reported ligands, leading to a unique coordination within the crystals (Supplementary Table 2).

To explore the unique role of the strong coordination between 3AP and the Pb–I inorganic framework, density functional theory (DFT) calculations were conducted for a super cell of an α-FAPbI₃ perovskite slab containing 3AP and other commonly used large organic ligands, such as 2-phenethylammonium (PEA), n-butylammonium (BA), 3-(aminomethyl)piperidinium (3AMP), and 3-(aminomethyl)pyridinium (3AMPY), on its surface (Fig. 1b and Supplementary Fig. 4). We calculated the adsorption energies of these ligands on the perovskite surface, formation energies of I⁻ vacancies in the perovskite lattice, and migration energy barrier of I⁻ vacancies (the calculation details are provided in density functional theory (DFT) calculations). The adsorption energy of 3AP (3.135 eV) was much higher than those of other perovskite systems containing 3AMP (2.784 eV), 3AMPY (2.988 eV), BA (2.968 eV), and PEA (2.498 eV) (Fig. 1c) due to the strong 3AP–perovskite coordination. The formation energy of I⁻ vacancies in the perovskite lattice increased from 1.075 eV for MAPbI₃ to 1.422 eV for α-FAPbI₃, which is consistent with the higher degree of hydrogen bonding in the second structure[31]. The formation energy of I⁻ vacancies further increased above 3.812 eV for the 3AP-perovskite system, which exceeded the values obtained for the perovskite systems with 3AMP (3.574 eV), 3AMPY (3.791 eV), BA (3.716 eV), and PEA (3.584 eV) ligands. Furthermore, the migration barrier energy of I⁻ vacancies increased from 0.737 to 1.467 eV after the addition of 3AP molecules, which is also higher than those of the systems containing other ligands (0.86–1.29 eV) (Supplementary Fig. 5 and Table 3). These results indicate that the strong 3AP–perovskite coordination plays the role in anion fixation and suppressing the migration of anion vacancies.

The coordination between 3AP molecules and Pb–I inorganic frameworks was further investigated by nuclear magnetic resonance (NMR) and X-ray photoelectron spectroscopy (XPS). Samples for analysis were prepared by dissolving neat 3AP or 3AP + PbI₂ (1:1) powder in d₆-dimethyl sulfoxide (DMSO) solvent. The ¹H resonance signals at 9.60 and 9.37 ppm originated from the C=NH₂⁺ and C–NH₂ environments of the amidino group of neat 3AP were shifted to 9.41 and 9.00 ppm for the 3AP + PbI₂ system (Fig. 1d)[32–34], respectively. The ¹H resonance signal caused by the pyridine ring was also slightly shifted to higher frequencies. These results suggest the formation of hydrogen bonds between 3AP and I⁻ [35,36]. The interactions between 3AP with PbI₂ were further confirmed by Fourier-transform infrared spectroscopy (FTIR) (Supplementary Fig. 6). We found that the stretching and bending vibration frequencies (marked as "s" and "b") of the –NH (s) and –NH (b) modes of the amidino group were shifted from 3312 to 3221 cm⁻¹ and from 1512 to 1523 cm⁻¹ in the presence of 3AP–PbI₂ hydrogen bonds, respectively. Correspondingly, the –CH (s) and –CH (b) modes of the pyridine ring[37] were shifted from 3130/3026 to 3080/2959 cm⁻¹ and from 760 to 771 cm⁻¹, respectively. Overall, these observations confirm the formation of strong hydrogen bonds between 3AP and the Pb–I inorganic framework, which is expected to stabilize I⁻ anions on the crystal surface or at the grain boundaries. Figure 1e shows the Pb 4f XPS profiles of neat PbI₂ and 3AP + PbI₂ compounds. The Pb 4f₅/₂ and Pb 4f₇/₂ peaks of PbI₂ located at 142.09 and 138.25 eV, respectively, were shifted towards lower binding energies by 0.4 eV, indicating the existence of an electron-rich environment of Pb in the 3AP-containing PbI₂ lattice due to the strong coordination between 3AP and the Pb–I inorganic framework[38–40].

The influence of the molecule–perovskite coordination on the activation energy (Eₐ) of ion migration was further experimentally

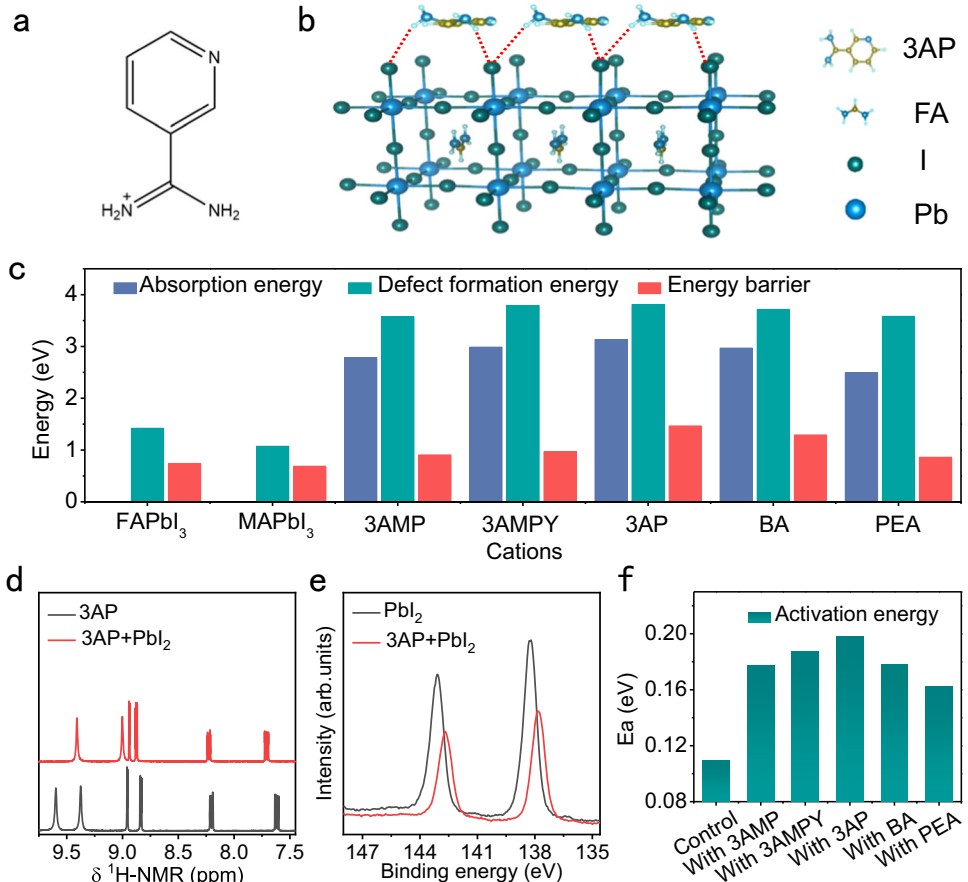

**Fig. 1 | 3AP–perovskite coordination. a** Chemical structure of the 3AP$^+$ ion. **b** Supercell constructed for DFT calculations, in which 3AP molecules are adsorbed on the FAPbI$_3$ (001) surface via FAI terminal groups. **c** Absorption energies of ligands on the perovskite surface, formation energies of I$^-$ vacancies in the perovskite structure, and migration energy barriers of I$^-$ vacancies in the perovskite lattice calculated for the perovskite systems with 3AMP, 3AMPY, 3AP, BA, and PEA ligands. **d** $^1$H-NMR spectra of neat 3AP and 3AP + PbI$_2$ compounds. **e** Pb 4f XPS profiles of the neat 3AP and 3AP + PbI$_2$ powders. **f** Activation energy (Ea) of ion migration derived from the temperature-dependent conductivities of perovskites with various ligands (see Supplementary Fig. 8). Source data are provided as a Source Data file.

evaluated by measuring the temperature-dependent conductivity of perovskite films (the schematic diagram of device structure is shown in Supplementary Fig. 7). The plots of temperature-dependent conductivity for the neat perovskite film (denoted as the control) and perovskite films with the above-mentioned ligands are shown in Supplementary Fig. 8[41–43]. The E$_a$ values (the E$_a$ values is calculated by Eq. 3 in the "Methods" section) increased from 0.109 eV for the control perovskite sample to 0.198 eV for the perovskite with 3AP (Fig. 1f). The last value is also higher than those of the perovskites containing 3AMP (0.177 eV), 3AMPY (0.187 eV), BA (0.178 eV), and PEA (0.162 eV). Thus, the presence of 3AP in the Pb–I framework can inhibit the formation of I$^-$ vacancy defects in PSCs.

## Photovoltaic performance

We evaluated the photovoltaic performance of PSCs without (denoted as the control) and with $x$ mol% ($x$ = 4, 8) iodized 3-pyridinecarboximidamide (3API). The solar cells were fabricated with a planar n-i-p device architecture (Fig. 2a), in which compact TiO$_2$ (c-TiO$_2$) and 2,2′,7,7′-tetrakis(N,N-di-p-methoxyphenylamine)−9,9′-spirobifluorene (Spiro-OMeTAD) were used as electron-transporting and hole-transporting layers, respectively. The cell performance was considerably enhanced by 3API addition, and the 3AP-based cells demonstrated higher open-circuit voltage ($V_{OC}$), fill factor (FF), short-circuit current density ($J_{SC}$), and PCE values as compared with those of the control cell (see the photovoltaic parameters in Supplementary Table 4). The maximum efficiency was observed for the device with

4 mol% 3API. The control device exhibited a maximum PCE of 22.76% with a $J_{SC}$ of 24.94 mA cm$^{-2}$, $V_{OC}$ of 1.123 V, and FF of 0.81. Meanwhile, the 3AP-based cells achieved a maximum PCE of 25.3% with $V_{OC}$ of 1.181 V, $J_{SC}$ of 26.04 mA cm$^{-2}$, and FF of 82.21% (Fig. 2b, blue data points). Some cells were sent to an accredited photovoltaic test laboratory (National Institute of Metrology, NIM, China) for certification, and their certified quasi-steady-state efficiency was 24.8% (Fig. 2b, red data points; the certification details are presented in Supplementary Fig. 9). Typical current density−voltage ($J$−$V$) characteristics of the control and 3AP-based devices are shown in Fig. 2c, and the stabilized power outputs (SPOs) of the corresponding cells are presented in Fig. 2d. The external quantum efficiency (EQE) of the certified cell yielded an integrated $J_{SC}$ of 25.97 mA cm$^{-2}$ (Fig. 2e) with a negligible variation (-0.8%) from the measured $J_{SC}$, which is higher than that of the control sample (integrated $J_{SC}$ of 24.55 mA cm$^{-2}$). The relationship between $V_{OC}$ and light intensity is plotted in Fig. 2f. It is based on the following equation[27,44,45]:

$$V_{OC} = (nk_B T/q)\ln(I) + B, \qquad (1)$$

where k$_B$ is the Boltzmann constant, T is the temperature, q is the electric charge, B is the constant, and n is the ideality factor. The 3AP-based cell exhibited a smaller slope than that of the control (1.26 k$_B$T/q vs. 1.60 k$_B$T/q), which indicates less intense defect-assisted non-radiative recombination in the 3AP-containing perovskite layer.

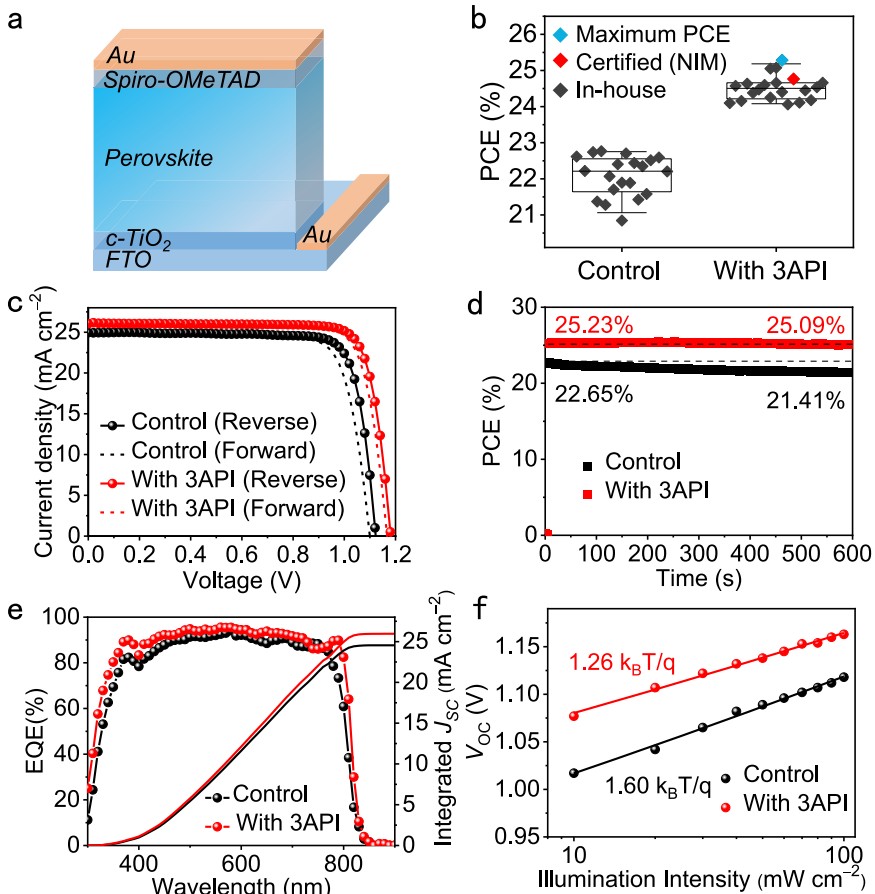

**Fig. 2 | Photovoltaic performance of the PSCs without (denoted as control) and with 3API. a** Planar n-i-p device structure. **b** Efficiency statistics of both solar cells including the certified value obtained at the National Institute of Metrology (NIM) and lab-measured efficiencies. **c** Current density–voltage (*J*–*V*) curves. **d** Stabilized power outputs (SPOs). **e** EQE curves and integrated current densities. **f** $V_{OC}$ dependences on light intensity. Source data are provided as a Source Data file.

## Film characterization

To investigate the effect of the molecule-perovskite coordination on perovskite properties, we performed grazing incidence wide-angle scattering (GIWAXS) characterization of the films. The films exhibited a predominant 3C (100) perovskite phase ($q = 1.02\,\text{Å}^{-1}$) with the $PbI_2$ ($q = 0.90\,\text{Å}^{-1}$) and undesired 6H (101) phases at $0.83\,\text{Å}^{-1}$ for the control sample (Fig. 3a). The 3API addition suppressed the formation of the undesired 6H phase but promoted the formation of the 3 C phase, as indicated by the higher diffraction intensity (Supplementary Fig. 10). This provides strong evidence that 3AP can result in a complete conversion from undesired phases to perovskite 3 C species, which is one of the key factors for the improvement of device performance.

Scanning electron microscopy (SEM) observations showed a slight increase in the average grain size of the perovskite film after 3API addition (1.4 vs. 1.0 µm) (Fig. 3b; see the statistical distribution in the inset). The larger grain size of the 3AP-based film is consistent with the behavior observed by GIWAXS. Monolithic grains were present in the cross-sectional SEM images from their tops to bottoms (Fig. 3c). Atomic force microscopy (AFM) observations revealed a decrease in the surface roughness from 23.3 to 17.0 nm with 3API addition due to the stitched grain boundaries (Supplementary Fig. 11). In addition, the Time-of-Flight Secondary Ion Mass Spectrometry (TOF-SIMS) was performed, which indicates a uniform distribution of 3AP$^+$ throughout the film as shown in Supplementary Fig. 12).

Steady-state and time-resolved optical microscopies were performed to explore the optoelectronic properties of the perovskite films. The 3API addition did not change the absorption threshold or photoluminescence (PL) peak position (Fig. 3d). The bandgaps of both films determined from their Tauc plots were equal to 1.54 eV (Supplementary Fig. 13). However, we observed an apparent increase in the PL intensity of the 3AP-based films, suggesting that 3API addition reduced the non-radiative recombination in the perovskite film, which was further validated by time-resolved photoluminescence (TRPL). The 3AP-based perovskite film demonstrated a full photoluminescence decay of 5.5 µs (Fig. 3e), which was much longer than the control (1.2 µs). Hence, 3AP addition suppressed the non-radiative carrier recombination, which was consistent with the results of DFT calculations indicating that the formation of I$^-$ vacancy defects was inhibited by the strong molecule–perovskite coordination, and accounted for the improved photovoltaic metrics.

We further performed space charge-limited current (SCLC) measurements to quantitatively estimate the defect density ($N_{defects}$) and charge mobility ($\mu_{electron}$) of the films. For this purpose, electron-dominated devices (fluorine-doped tin oxide (FTO)/c-TiO$_2$/perovskite/[6,6]-phenyl-C61-butyric acid methyl ester (PCBM)/Au) were fabricated and characterized under different biases in the dark (Fig. 3f). The $N_{defects}$ and $\mu_{electron}$ are calculated by Eqs. 4 and 5, respectively. Which are described in the Material characterization. The value of $N_{defects}$ decreased from $1.43 \times 10^{16}$ to $4.76 \times 10^{15}\,\text{cm}^{-3}$ with 3API addition, while the value of $\mu_{electron}$ increased from 0.139 to 0.339 cm$^2$ V$^{-1}$ s$^{-1}$. This result is consistent with the suppressed non-radiative carrier recombination probed by TRPL and attributed to the strong 3AP–perovskite coordination that suppressed I$^-$ vacancy formation. Capacitance–voltage (C–V) measurements were also performed to elucidate the higher $V_{OC}$ values. The obtained C$^{-2}$–V plots and built-in potentials ($V_{bi}$) are presented in Supplementary Fig. 14. It shows that

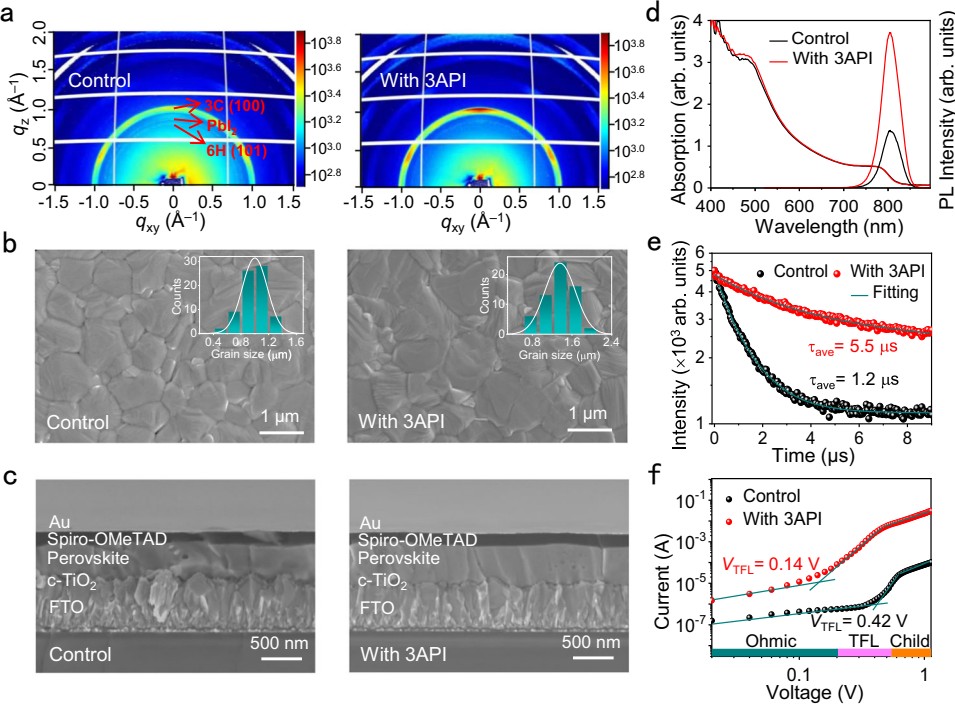

**Fig. 3 | Characteristics of the control and 3AP-based perovskite films. a** Two-dimensional GWIAXS images of the final two films obtained after thermal annealing. **b** SEM images of the final films. The insets show the grain size distributions. **c** The cross-sectional SEM images of the final films. **d** Steady-state UV–Vis absorption and PL spectra of the films. **e** TRPL intensities of the films. 'average' is denoted as 'ave'. **f** Dark J–V curves of the electron-only devices with the FTO/c-TiO₂/perovskite/ PCBM/Ag architecture. $V_{TFL}$ is the trap-filled limit voltage. Source data are provided as a Source Data file.

value of $V_{bi}$ increased from 1.108 to 1.160 V with 3API addition, which is similar to the $V_{OC}$ trend obtained from the J–V curves and suggests the existence of a wider depletion region for the 3AP-based device[28]. Thus, the charge separation and carrier transport are significantly improved after 3API addition.

The ultraviolet photo-electron spectroscopy (UPS) test was employed to further illustrate the effect of 3API on the energy band alignment of perovskite (Supplementary Figs. 15 and 16). The introduction of 3API reduced the VBM and CBM of perovskite from −5.46 eV and −3.92 eV to −5.55 eV and −4.01 eV, respectively. The results indicate better energy level alignment between the perovskite layer and the electron transport layer, which is beneficial for charge extraction and transfer. Transient photocurrent (TPC) and transient photovoltage (TPV) were also performed to confirm the effectiveness of 3AP⁺ (Supplementary Fig. 17a, b). Compared to the reference device, the device with 3API exhibited shorter charge extraction time (4.25 vs. 9.38 μs) and longer charge compounding time (11.2 vs. 3.07 μs). This indicates that the introduction of 3API can effectively improve the charge extraction process and suppress charge complexation, which is consistent with the UPS results. Kelvin probe force microscopy (KPFM) analysis indicates a higher electronic chemical potential of perovskite with the 3AP addition (Supplementary Fig. 18). The higher electronic chemical potential and reduced energy level (Supplementary Figs. 15 and 16) indicate a less n-type surface, which can facilitate the hole extraction in devices[19].

Furthermore, we investigated the effects of different halides (3APX, X = Cl, Br) and molecular stacking on the formation of I⁻ vacancies in perovskites. First, I⁻ ions were replaced with chloride anions (Cl⁻) and bromide anions (Br⁻) while preserving the 3AP⁺ ions and replicated the similar observations with 3API addition. The resulting films exhibited enlarged grain size (see the SEM images in Supplementary Fig. 19), decreased film surface roughness (Supplementary Fig. 20), increased PL intensity (Supplementary Fig. 21), decreased nonradiative recombination decay (Supplementary Fig. 22),

and decreased trap densities (Supplementary Fig. 23) as compared with those of the control. The PCEs of the devices containing 3APCl (24.6%) and 3APBr (24.7%) are comparable to that of the 3API-based cell (see the J–V curves in Supplementary Fig. 24 and photovoltaic parameters in Supplementary Table 5), suggesting that 3AP⁺ ions are responsible for eliminating vacancy defects. Thus, strong molecule−perovskite coordination is required to suppress the formation of vacancy defects. In addition, we also verify the positive role of 3AP in other different perovskite compositions. Both $FA_{0.9}Cs_{0.1}PbI_3$ and $FA_{0.92}MA_{0.08}PbI_3$ devices exhibit effective enhancements of photovoltaic parameters with the 3API addition (Supplementary Figs. 25 and 26), which demonstrates the universality of the 3AP additive.

**Stacking defects**

In sought to better understand how the 3AP addition influences stacking defects of perovskites, in situ PL spectroscopy and photocurrent mapping were performed during perovskite fabrication process and for devices when working under illumination, respectively. Figure 4a, b shows the in situ PL spectra of the films during spinning and thermal annealing. During spinning, both films exhibit negligible emission (Fig. 4a, b, left). During the annealing process (Fig. 4a, b, right), the suggesting no illuminating α-FAPbI₃ phase shows a rapid formation as evidenced from a quick rise of PL intensity. The following sharp decrease of PL intensity can be attributed to solvent volatilization, film crystallization restructuring, and negative temperature coefficient[46–48]. Note that the control film exhibits a continuous but a slow decrease of PL intensity with further thermal annealing (Fig. 4c), which is indicative of the formation of stacking defects. The 3API addition not only retarded the perovskite crystallization process but also inhibited the formation of stacking defects, as confirmed by the slowly increased PL intensity with further thermal annealing (Fig. 4d). This indicates that in situ crystallization modulation and defect passivation can be realized through the strong anchoring effect of 3API with $[PbI_6]^{4-}$ during the formation of the films.

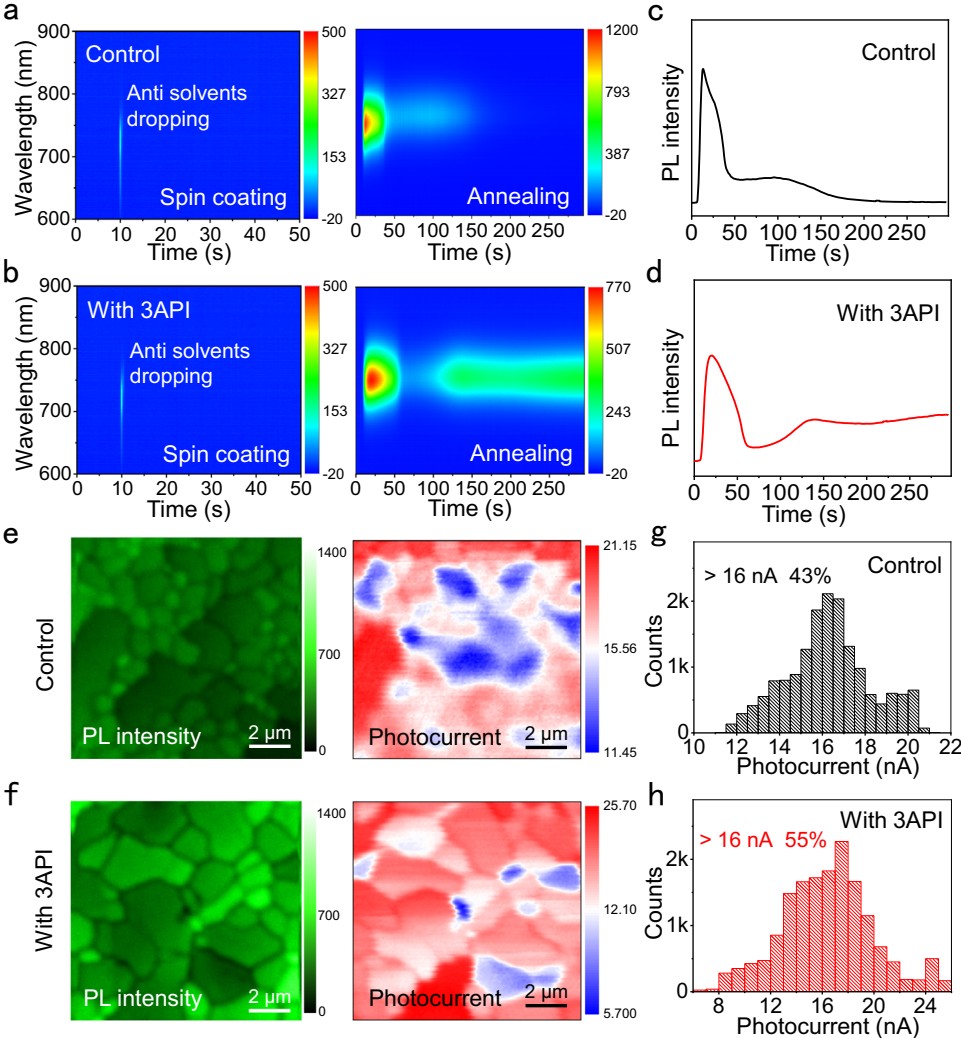

**Fig. 4 | Stacking defects during film formation and in the working devices.**
**a, b** In situ PL spectra of films with or without 3API during spin coating and
annealing. **c, d** The PL intensity extract from **a** and **b**. **e, f** PL intensity and
photocurrents of the working devices with or without 3API. **g, h** The histogram of
photocurrent statistics. Source data are provided as a Source Data file.

The PL intensity and photocurrent mappings on the working
PSCs were recorded by using a laser-scanned and time-resolved PL
microscopy coupled with a photocurrent detection module as we
reported previously[49]. We collected the PL intensity and local pho-
tocurrent mapping within a $10.5 \times 10.5\,\mu m$ area for both devices
(Supplementary Fig. 27). Figure 4e, f, left shows the PL intensity
mapping, where the distribution of grain size and shape is clearly
identifiable. The devices with 3API show a larger perovskite grain
size (Supplementary Fig. 28a) and are coupled with a stronger PL
intensity than the control devices, which is consistent with the SEM
(Fig. 3b), PL (Fig. 3d), TRPL (Fig. 3e) and PL mapping (Supplemen-
tary Fig. 21) results. Furthermore, we observed fewer photons from
larger grains for both devices, which indicates more efficient charge
separation and extraction. The photocurrent mapping on the same
area are shown in Fig. 4e, f, right. The devices with 3API exhibit
higher photocurrents compared to the control (Supplementary
Fig. 28b), which further indicates less stacking defects with 3API. We
also performed statistical histograms for the different photocurrent
values in the photocurrent images. It is clear that the percentage of
photocurrent values >16 nA in the devices increase from 43 to 55%
after the introduction of 3API (Fig. 4g, h). Both in situ observation
during film formation and ex situ analysis on the working devices
explain well the importance of in situ passivation of stacking defects

during the formation for highly efficient charge separation and
extraction in the working devices.

## Stability enhancement of solar cells

To assess the stability of solar cells, we first measured the intrinsic
phase stability of perovskite films stored in a dark air environment at a
temperature of 25 °C and relative humidity of 30–40%. Figure 5a shows
the X-ray diffraction (XRD) patterns of the control and 3AP-based
perovskite films recorded before and after ambient aging for 74 days.
The diffraction signals of the $\delta$-FAPbI$_3$ and PbI$_2$ phases became more
prominent in the control perovskite film after aging. However, these
undesired phases were not detected in the 3AP-based perovskite film
stored under the same conditions, indicating that the strong
molecule–perovskite coordination can inhibit the transformation and
degradation of the 3C phase. The cross-sectional SEM images and
corresponding Energy dispersive spectroscopy (EDS) of complete cells
before and after aging indicate less I⁻ migration toward Spiro-OMeTAD
layer (Supplementary Fig. 29) with the 3AP addition, which is due to
strong hydrogen bonding and anchoring effect of 3AP on perovskite.
We further investigated the thermal stability of unencapsulated solar
cells stored at 85 °C under a nitrogen atmosphere (Fig. 5b). We used
Poly[bis(4-phenyl)(2,4,6-trimenthylphenyl)amine] (PTAA) instead of
Spiro-OMeTAD and the organic passivation layer was removed to

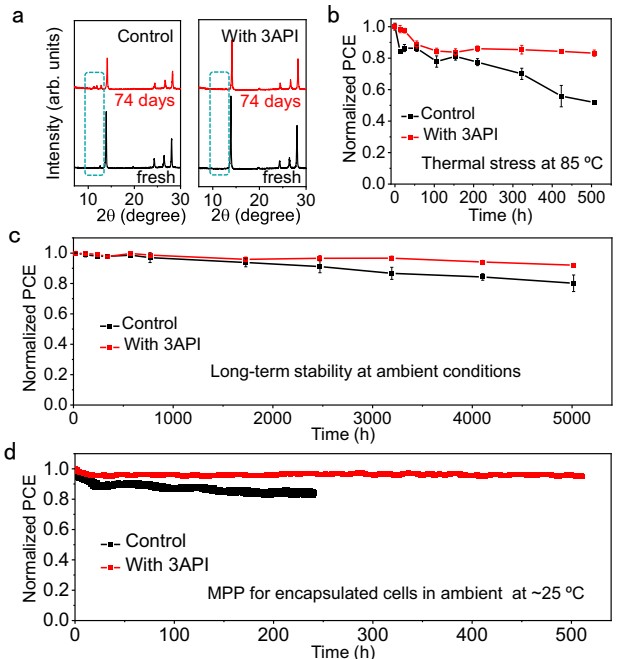

**Fig. 5 | Stability studies of perovskite films and solar cells. a** XRD patterns of the control and 3AP-based perovskite films recorded before and after aging for 74 days. **b** PCE evolution observed for the control and 3AP-based devices stored without encapsulation at 85 °C under a nitrogen atmosphere. **c** PCE evolution observed for the control and 3AP-based devices stored without encapsulation at ambient conditions (25 °C and 30–40% relative humidity). **d** Stabilized maximum power point (MPP) values recorded for the control and 3AP-based devices with encapsulation under continuous light irradiation with a white LED lamp at 100 mW cm⁻² in ambient (25 °C and 30–40% relative humidity). All the error bars represent the standard deviation for devices. Source data are provided as a Source Data file.

avoid introducing sources of instability. The 3AP-based cell retained 83% of its initial efficiency after 500 h of thermal treatment, which was higher than that of the control cell (52% of its initial efficiency). Furthermore, the long-term environmental stability of the unencapsulated cells stored under ambient conditions (25 °C and 30–40% relative humidity) was also investigated (Fig. 5c). The PCEs decreased by ~20% for the control cell and only by 8% for the 3AP-based cell after 5000 h. Compared with the control cell, the 3AP-based cell exhibits a smaller FF loss during aging (Supplementary Fig. 30 and Table 6). Finally, we checked the operational stability of encapsulated solar cells by performing maximum power point (MPP) tracking in the ambient (25 °C and 30–40% relative humidity) following the ISOS-L-1 stability protocol[50]. The 3AP-based solar cells exhibits a 5% decrease of their initial efficiency after 510 h. While the PCE of the control device decreased by 16% after only 240 h (Fig. 5d). From these observations, we can conclude that the anion-vacancy defect engineering via strong molecule–perovskite coordination provides an effective and simple solution for increasing both the efficiency and stability of PSCs.

## Discussion

In this study, we demonstrate that anion fixation and associated undercoordinated-Pb passivation in FAPbI₃-based perovskites can be realized synergistically by functionalizing perovskite surface with amidino-based ligand 3AP. The results of theoretical calculations combined with the NMR, XPS, in situ PL, and in-depth optoelectronic characterization data helped elucidate the mechanism of the molecule–perovskite coordination and its effect on the anion fixation and vacancy defect formation in the perovskite structure. The anion-vacancy defect engineering via the molecule–perovskite coordination increased the PCE of planar PSCs from 22.8 to 25.3% (certified at 24.8%)

and their stability. Our findings offer an effective chemical strategy for the facile fabrication of high-performance stable PSCs and are potentially applicable to other perovskite optoelectronic devices.

## Methods

### Materials

Iodized 3-pyridinecarboximidamide (3API) was synthesized by dissolving 1 g of pyridecarboximidamide (98%, Damas-beta) and 2.25 g hydroiodic acid (HI, 57 wt.% in water, Aldrich) in 10 mL ethanol followed by stirring at 0 °C for 6 h for crystallization. The obtained 3API crystals was dissolved in ethanol for recrystallization. The resulting product was dried at 50 °C in a vacuum oven for 24 h. 3APBr and 3APCl were obtained via a similar fabrication procedure.

Formamidinium iodide (FAI), Pb (II) iodide (PbI₂), and spiro-OMeTAD were procured from Advanced Election Technology Co., Ltd. Methylammonium chloride (MACl), Poly[bis(4-phenyl)(2,4,6-trimethylphenyl)amine]) (PTAA) and phenylethylammonium iodide (PEAI) were obtained from Xi'an Polymer Light Technology Co. Chlorobenzene (CB) (anhydrous, 99.8%), Toluene (anhydrous, 99.8%) and isopropanol (IPA) (anhydrous, 99.5%) were purchased from Sinopharm Chemical Reagent Co., Ltd. N,N-dimethylformamide (DMF) (anhydrous, 99.8%) and dimethyl sulfoxide (DMSO) (anhydrous, ≥99.9%) were obtained from Alfa Aesar Inc. 4-tert-butylpyridine (t-BP, 99%), bis(trifluoromethane)sulfonamide lithium salt (Li-TFSI, 99.95% trace metals basis), HI solution (55–58 wt.% in water), HBr solution (48 wt.% in water), and HCl (37 wt.% in water), were purchased from Aladdin.

### Synthesis of (3AP)PbI₄ single crystals.

The synthesis and characterization of (3AP)PbI₄ single crystals can be found in previous work[30]. Briefly, 3 mmol (670 mg) PbO was first dissolved into a mixtures solution including 30 mL of HI solution and 3 mL of H₃PO₂ solution (50 wt.% in water, Aladdin), and stirred continuously at 180 °C to obtain a clear solution. Then, 3 mmol (473 mg) of 3APCl was added to the above solution, and stirred continuously at 180 °C until 3APCl was completely dissolved. The temperature of the above solution was then lowered to room temperature to allow the crystals to precipitate. Finally, the products were filtered and dried in an oven at 50 °C for 12 h. All the processes were operated in ambient air.

### Perovskite film preparation and device fabrication

Perovskite solar cells (PSCs) were fabricated with a planar n-i-p structure of fluorine-doped tin oxide (FTO)/c-TiO₂/FAPbI₃/Spiro-OMeTAD/Au. FTO glass substrates (2.5 × 2.5 cm) were cleaned by ultrasonication in a special cleaning concentrate (Hellmanex III (Sigma–Aldrich) mixed with ultrapure water at a volume ratio of 1–1.5:100) and ultrapure water for 20 min. After that, they were cleaned sequentially with IPA and ethanol for 20 min. Finally, the substrates were dried under a nitrogen flow and treated with ultraviolet (UV) light for 15 min before the deposition of c-TiO₂. c-TiO₂ layers were prepared by chemical bath deposition on FTO substrates. Before spin coating, FTO/c-TiO₂ substrates were annealed at 200 °C for 30 min in air and treated with UV-ozone for 15 min. To fabricate the FAPbI₃ perovskite thin film, a precursor solution (1.5 M) was prepared by dissolving PbI₂ and FAI in a solvent mixture of DMF and DMSO (volume ratio: 4:1). 3APX (X = I, Br, Cl) were added to the precursor solutions at different concentrations ranging from 0 to 8 mol%. MACl (34.4 mg) was added to all perovskite precursor solutions. For each sample, 70 μL of the filtered solution was spread over the FTO/c-TiO₂ substrate at 6000 rpm for 50 s. During spin coating, 1 mL of diethyl ether was dripped after spinning for 10 s using a pipette. The film was annealed on a hot plate at 150 °C for 10 min to obtain a black film, and after cooling to room temperature (~25 °C), the perovskite film was spin-coated with PEAI dissolved in IPA (20 mM) at 3000 rpm for 30 s without further processing. The hole-transporting layer was deposited onto the top of the perovskite layer

at a spin rate of 5000 rpm for 30 s using a Spiro-OMeTAD solution, which consisted of 90 mg Spiro-OMeTAD, 36 μL t-BP, 22 μL Li-TFSI (520 mg Li-TFSI in 1 mL acetonitrile), and 1 mL CB. For the thermal stability of devices (with an initial PCE of 21.59%, a $V_{OC}$ of 1.104 V, a $J_{SC}$ of 25.43 mA cm$^{-2}$, a FF of 76.89% for With 3API, and an initial PCE of 19.45%, a $V_{OC}$ of 1.063 V, a $J_{SC}$ of 25.21 mA cm$^{-2}$, a FF of 72.62% for Control), 12 mg PTAA was dissolved in 1 ml toluene, then 6 μL Li-TFSI (340 mg Li-TFSI in 1 mL acetonitrile) and 6 μL tBP were added. The PTAA solution was spin-coated on surface of the perovskite layer at 2000 rpm. for 30 s. Finally, an 80-nm Au electrode was vapor-deposited on the top of the Spiro-OMeTAD layer, and the fabricated device was oxidized for 24 h.

## Device characterization

$J$–$V$ characteristics of the fabricated devices were obtained using a Keithley 2400 SourceMeter under ambient conditions at room temperature. Both reverse scans (1.7 → 0 V, step: 0.02 V, delay time: 20 ms) and forward scans (0 → 1.7 V, step: 0.02 V, delay time: 20 ms) were conducted. The area of the cell is 0.09 cm$^2$ and a mask of 0.07274 cm$^2$ (certificated by NIM, China. The certificate No.: CDjc2021-11025) was used to determine the effective area of the device before the test. The power output of the lamp was calibrated using a National Renewable Energy Laboratory traceable KG5-filtered silicon reference cell. The external quantum efficiency (EQE) was measured on a QE-R system (Enli Technology Co., Ltd.) using a 300-W Xe lamp as the light source. The monochromatic light intensity for EQE measurements was calibrated using a reference silicon photodiode. The maximum power point (MPP) tracking were obtained using solar cell aging test system (Shenzhen Lancheng Technology Co., Ltd.) under a simulated continuous AM1.5 G illumination (LED, 100 mW cm$^{-2}$, encapsulation devices in ambient at 25 °C and 30–40% relative humidity).

## Material characterization

UV-visible absorption spectra were acquired on a PerkinElmer UV-Lambda 950 instrument. Steady-state photoluminescence (PL) and time-resolved photoluminescence (TRPL) spectra were recorded with a PicoQuant FT-300 spectrometer. The excitation laser wavelength is 510 nm, the frequency is 40 MHz (for PL) and 0.5 MHz (for TRPL), and the detection wavelength range is 500–900 nm. X-ray diffraction (XRD) studies were performed using a DX-2700BH diffractometer (Dandong Haoyuan Instrument Co., Ltd.). Grazing incidence wide-angle scattering (GIWAXS, $\lambda = 1.24$ Å, incidence angle: 0.40°) was performed at the National Facility for Protein Science in Shanghai. Scanning electron microscopy (SEM) images were obtained using a Hitachi SU-8020 field-emission scanning electron microscope (Japan). Atomic force microscopy (AFM) images were obtained by a Bruker Dimension Icon instrument (USA). X-ray photoelectron spectroscopy (XPS) was performed on a photoelectron spectrometer (ESCALAB Xi+, Thermo Fisher Scientific). Fourier transform infrared (FTIR) spectra were recorded with a Bruker VERTEX 70 infrared spectrophotometer using KBr sheets. $^1$H nuclear magnetic resonance spectra were recorded on a JNM–ECZ400R/S1 apparatus (JEOL, Japan). Capacitance–voltage (C–V) measurements were performed on an electrochemical workstation (Modulab XM, USA) and the frequency was set to 200 kHz and the scan voltage range was 0–1.6 V. The built-in potential ($V_{bi}$) was further calculated according to the Mott–Schottky equation:

$$C^{-2} = 2(V_{bi} - V_a)/A^2 e\varepsilon\varepsilon_0 N_A \qquad (2)$$

where $N_A$ and $V_a$ are the carrier concentration and applied voltage, respectively; $\varepsilon_0$ and $\varepsilon$ denote the vacuum permittivity and dielectric constant of the perovskite, respectively; and e represents the electron charge[28,33].

Temperature-dependent conductivity measurements were performed using a Keysight 4200 source meter to determine the ion

migration activation energies ($E_a$) of perovskite films at different temperatures (188–333 K)[43], were placed in a manual dark probe station equipped with a temperature controller to obtain a required temperature, and the sweep voltage range was 0–3 V. The value of $E_a$ was further calculated according to the Nernst–Einstein relation[41,42]

$$\sigma T = \sigma_0 \exp(-E_a/k_B T) \qquad (3)$$

where $\sigma_0$ is the constant, $k_B$ is the Boltzmann constant, and T is the temperature (K).

Defect density ($N_{defects}$) and charge mobility ($\mu_{electron}$) were determined from the electron-only dark $J$–$V$ curves obtained by a space charge-limited current (SCLC) method. The device architecture was FTO/c-TiO$_2$/perovskite/PCBM/Ag. $N_{defects}$ was calculated according to the equation[51]:

$$N_{defects} = (2\varepsilon\varepsilon_0 V_{TFL})/(eL^2) \qquad (4)$$

where $\varepsilon_0$ and $\varepsilon$ denote the vacuum permittivity and dielectric constant of the perovskite, and e and L represent the electron charge and thickness of the perovskite film, respectively. $\mu_{electron}$ was computed according to the Mott–Gurney law[52]:

$$\mu_{electron} = (8L^3 K)/(9\varepsilon\varepsilon_0) \qquad (5)$$

where K denotes the slope of the Child regime.

## DFT calculations

First-principles calculations were performed in the framework of the density functional theory (DFT) using the program package CASTEP, plane-wave ultra-soft pseudopotential (PW–USPP) method, and Perdew–Burke–Ernzerhof (PBE) form of the generalized-gradient approximation (GGA) exchange–correlation energy functional. Structural optimizations of FAPbI$_3$ and MAPbI$_3$ were conducted using the Broyden–Fletcher–Goldfarb–Shanno (BFGS) algorithm by allowing all atomic positions to relax. To calculate the total energies of 3AMP, 3AMPY, 3AP, BA, and PEA ions, they were placed inside a 10 Å × 10 Å × 10 Å lattice while allowing their atomic positions to vary. The simulations stopped when the total energies converged to 10$^{-5}$ eV/atom, forces on each unconstrained atom were smaller than 0.03 eV/Å, stresses were lower than 0.05 GPa, and displacements were <0.001 Å. The plane-wave cutoff, $E_{cut}$, was set to 340 eV. K-point meshes of 4 × 4 × 4, 3 × 3 × 2, and 1 × 1 × 1 were used for the Brillouin zone sampling of FAPbI$_3$, MAPbI$_3$, 3AMP, 3AMPY, 3AP, BA, and PEA ions.

**Absorption energy.** To calculate the adsorption energies of 3AMP, 3AMPY, 3AP, BA, and PEA ligands on the FAI-terminated FAPbI$_3$ (001) surface, we modeled the first five layers of this surface. To obtain the most stable structure, structural optimization was performed by fixing the bottom three layers and setting the thickness to 15 Å vacuum. Adsorption energies were calculated by the following equation:

$$\Delta E_{ad} = E_{adsorption\_system} - E_{FAPbI3\_001} - E_t[adsorbate] \qquad (6)$$

where $E_{adsorption\_system}$, $E_{FAPbI3\_001}$, and $E_t[adsorbate]$ represent the total energies of the adsorption system, FAI-terminated FAPbI$_3$ (001) surface, and adsorbate, respectively.

**Defect formation energy.** The formation energy of I$^-$ vacancy defects in the perovskites with 3AMP, 3AMPY, 3AP, BA, and PEA ligands on the FAI-terminated FAPbI$_3$ (001) surface was calculated by the following equation:

$$\Delta E_{defect} = E_{defect\_system} + E_t[I] - E_{perfect\_system} \qquad (7)$$

where $E_{\text{defect\_system}}$, $E_{\text{perfect\_system}}$, and $E_t[\text{I}]$ represent the total energies of the defect system, the perfect system, and a single I atom, respectively.

**Transition state (TS).** To determine the migration energy barriers of I⁻ ions on the FAI-terminated FAPbI₃ (001) and MAI-terminated MAPbI₃ (001) surfaces, we performed TS searches using the complete linear synchronous transit/quadratic synchronous transit method.

**Reporting summary**

Further information on research design is available in the Nature Portfolio Reporting Summary linked to this article.

## Data availability

The data that support the findings of this study are available in the Supplementary Information/Source Data file. Source data are provided with this paper.

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

## Acknowledgements

This work was supported by National Natural Science Foundation of China (61974085, K.Z.), the 111 Project (B21005, S.L.), the National 1000 Talents Plan program (1110010341, S.L.), National University Research Fund (GK202201005, K.Z.). The authors also thank beamlines BL17B1 and BL14B1 at the SSRF for providing the beamtime and User Experiment Assist System of SSRF for their help. We also thank Prof. Jingbi You for the useful discussions related to the fabrication of perovskite films.

## Author contributions

T.Y. and L.G. conducted the most experimental and DFT studies. J.L. assisted with the PL mapping procedure and C–V and FTIR measurements. C.M. helped with the synthesis and characterization of 3APX (X = I, Br, Cl) single crystals. Y.D., D.L., and Y.Y. assisted with the GIWAXS acquisition. P.W. assisted with the PL and TRPL measurements. Z.D. helped fit the $V_{OC}$–Sun data. J.F. and S.W. assisted with the XRD measurements. P.X. and W.T. assisted with the PL intensity and photocurrent mapping measurements. H.L. participated in the discussion of the obtained results. X.C. helped with the J–V measurements. K.Z. conceived the study. K.Z. and S.L. supervised the study. All authors discussed the results and commented on the manuscript.

## Competing interests

The authors declare no competing interests.
