## [Peer Review File · Nature Communications]

One-stone-for-two-birds strategy to attain beyond 25% perovskite solar cellsREVIEWER COMMENTS

Reviewer #1 (Remarks to the Author):

In this work, the authors present a named one-stone-for-many-birds strategy in which both anion-fixation and associated undercoordinated-Pb passivation are achieved by using a single amidino-based ligand, namely 3-amidinopyridine (3AP), which enabled the devices with an excellent power conversion efficiency (PCE) of 25.3% (certified at 24.8%) and ambient stability. However, the PCE of the target devices declined to 90% of initial PCE after merely 200 h aging through maximum power point (MPP) tracking under simulated AM1.5G illumination. To the best of my knowledge, the operational stability is far inferior to the recent reported results (e.g. *Science*, 2021, 373, 561–567.; *Energy Environ. Sci.*, 2021,14, 5552-5562; *Adv. Funct. Mater.* 2022, 2201193.; *Nat Energy* 2022, 7, 144-152.). Therefore, I think the marvelous stability the authors refer to is not appropriate. Moreover, the mechanism investigation and characterization methods are not thorough enough and I cannot recommend this work for publication in the current version.

Some comments:

1. This work demonstrates the effectiveness of 3-AP additives in fabricating highly efficient FAPbI₃ perovskite solar cells. However, can 3-AP play a positive role in the fabrication of other perovskite compositions or the inverted devices? Is the 3-AP additive strategy applicable to the fabrication of large-scale devices or modules? Please add relevant experiments to elucidate the universality of the 3-AP additives.
2. The existence form and distribution of 3-AP molecules is not clear.
 - a) The authors argued that 3-AP additives play both role in anion fixation and suppressing the migration of anion vacancies on the crystal surface or at the grain boundaries. Please provide proof of the spatial distribution of 3-AP molecules in the perovskite film, such as Nano-IR or TOF-SIMS.
 - b) The amidino-based spacers have been widely studied for 2D Ruddlesden–Popper (RP) perovskites (*Adv. Energy Mater.* 2020, 2000694.; *ACS Energy Lett.* 2021, 6, 2072-2080). The authors have proved that 3AP molecules were arranged in parallel and anti-symmetrically between Pb-I frameworks. Hence, could 3-AP additives form the low-dimensional phase in this work?
 - c) The detailed preparation method and characterization of (3AP)PbI₄ single crystals should be included in the manuscript or supporting information.
3. The stability test is insufficient to elucidate the marvelous stability.
 - a) For the operational stability test, in page 9 line 45, the word “approximately 30°C” is not scientific. Moreover, operational stability tests with the longer aging time under MPP tracking should be provided, such as 1000 h.

b) The authors refer to that 3-AP molecules realized anion-fixation and undercoordinated-Pb passivation in the form of hydrogen bonding. As we all know, the hydrogen bonding is much weaker than coordination and ionic bonding. Moreover, the 3-AP is a hydrophilic molecule. How can you achieve such a huge increase in ambient stability?

4. The authors illuminated that 3-AP molecules could form strong chemical bonds with the Pb-I framework. Since Pb and I determine the VB and CB of perovskite, 3-AP might influence the band alignment of perovskite film, which I think is one of the factors that enables the devices to achieve distinguished performance. The author should add the relevant experiments and discussions to the manuscript.

5. In page 7 line 35, the authors refer to “a decrease in the surface roughness from 23.3 to 17.0 nm with 3API addition due to the stitched grain boundaries”, however, the SEM images reveal that there exist many cracks and pinholes at the grain boundaries of 3-AP treated film. Please provide a reasonable explanation.

6. The cross-sectional SEM images of the fabricated perovskite solar cells should be provided. And it is recommended to conduct further characterizations of the devices to confirm the effectiveness of 3-AP molecules, such as TPC, TPV testing.

7. In the TRPL measurements, this expression of “was 5 times longer than that of the control (1.2 μ s) is incorrect.

Reviewer #2 (Remarks to the Author):

This paper reports the positive effects of 3AP molecule addition on PV performance and stability for PSCs. Characterization data and DFT calculation results clearly support the positive effects of 3API. The result offers a chemical strategy for efficient stable PSCs so it is recommendable for publication in Nature Communication after clearing the following concerns.

1. The main reason that authors insist on positive results is suppressing the formation of I- vacancy by adding 3API. However, in Figure 3a, the 3AP also influences the crystal orientation of FAPbI₃ that FAPbI₃ with 3API shows a more oriented (001) direction than the control film. The orientation or crystallinity of film can influence the final device performance in addition to the trap density of the film. So additional experiments for post-surface passivation treatment are recommended to exclude orientation and crystallinity of neat FAPbI₃ film. [<https://www.nature.com/articles/s41566-019-0398-2>, <https://pubs.rsc.org/en/content/articlehtml/2021/xx/c9ee00751b>]

2. The addition of 3API can change the work function of FAPbI₃ and result in an increase in V_{bi} . The UPS measurement for control film and FAPbI₃ film with 3API can support the result.

3. In Figure 4b, spiro-OMeTAD normally shows poor thermal stability at 85 °C due to its low glass transition temperature. So it is hard to compare thermal stability between PSCs with spiro-OMeTAD. For estimation of thermal stability between control and target films, other HTMs should be used such as P3HT and PTAA instead of spiro-OMeTAD.

4. The detail for PL, TRPL (ex, excitation fluence...), M-S analysis (ex. Frequency...), and temperature-dependent conductivity measurements (ex, device structure....) should be described more in the experiments part.

List of point-to-point response of reviewers' comments

Reviewer #1: In this work, the authors present a named one-stone-for-many-birds strategy in which both anion-fixation and associated undercoordinated-Pb passivation are achieved by using a single amidino-based ligand, namely 3-amidinopyridine (3AP), which enabled the devices with an excellent power conversion efficiency (PCE) of 25.3% (certified at 24.8%) and ambient stability. However, the PCE of the target devices declined to 90% of initial PCE after merely 200 h aging through maximum power point (MPP) tracking under simulated AM1.5G illumination. To the best of my knowledge, the operational stability is far inferior to the recent reported results (e.g. Science, 2021, 373, 561–567.; Energy Environ. Sci., 2021,14, 5552-5562; Adv. Funct. Mater. 2022, 2201193.; Nat Energy 2022, 7, 144-152.). Therefore, I think the marvelous stability the authors refer to is not appropriate. Moreover, the mechanism investigation and characterization methods are not thorough enough and I cannot recommend this work for publication in the current version.

Response: Thanks reviewer for the positive comments. We have done a series of additional experiments and referred to previous publications, with aim to significantly improve the manuscript based on the suggestions. The detailed responses to each point are shown below.

1. This work demonstrates the effectiveness of 3-AP additives in fabricating highly efficient FAPbI₃ perovskite solar cells. However, can 3-AP play a positive role in the fabrication of other perovskite compositions or the inverted devices? Is the 3-AP additive strategy applicable to the fabrication of large-scale devices or modules? Please add relevant experiments to elucidate the universality of the 3-AP additives.

Response: We appreciate the insightful comments on the universality of the 3AP additives. We have run additional experiments to verify the positive effects of 3AP in other different perovskite systems. In addition, we have also tried large area (1 cm²) device.

Figure R1 shows the *J-V* curve of FA_{0.9}CS_{0.1}PbI₃ device (n-i-p structure) with/without 3API. The 3API addition significantly increases open circuit voltage (*V*_{OC}) and fill factor (FF), resulting in PCE increase from 22.15% to 23.87%. **Figure R2** shows the *J-V* curve of FA_{0.92}MA_{0.08}PbI₃ device (n-i-p structure) with/without 3API. We also observe similar enhancement of *V*_{OC} and FF for this cell with 3API addition, with the efficiency increasing from 20.86% to 23.10%. We also tried wide bandgap Cs_{0.22}FA_{0.78}Pb(I_{0.85}Br_{0.15})₃ device (p-i-n structure, denote as CF) with/without 3API (**Figure R3**). The 3API addition increases PCE from 19.17% to 20.25%. These results indicate the universal role of 3API additive in passivating traps and device improvement.

Figure R1 The J - V curves of $\text{FA}_{0.9}\text{Cs}_{0.1}\text{PbI}_3$ device with/without 3API.

Figure R2 The J - V curves of $\text{FA}_{0.92}\text{MA}_{0.08}\text{PbI}_3$ device with/without 3API.

Figure R3 The J - V curves of $\text{Cs}_{0.22}\text{FA}_{0.78}\text{Pb}(\text{I}_{0.85}\text{Br}_{0.15})_3$ (CF) device with/without 3API.

The 3AP addition also helps improve large-scale fabrication of device. **Figure R4** presents the optical photographs and photovoltaic performance of 1-cm²-device. We also observed PCE enhancement from 18.30% to 20.41% with the 3API addition. Further improvement is expected to be obtained when well-controlling film uniformity during film formation.

Figure R4 The optical photograph and photovoltaic parameters of 1 cm² devices with/without 3API.

We have updated the manuscript by including additional description and Figures (**Supplementary Fig. 20** and **21**) as following (See **Page 9**):

*“In addition, we also verify the positive role of 3AP in other different perovskite compositions. Both FA_{0.9}CS_{0.1}PbI₃ and FA_{0.92}MA_{0.08}PbI₃ devices exhibit effective enhancements of photovoltaic parameters with the 3API addition (**Supplementary Fig. 20** and **21**), which demonstrates the universality of the 3AP additive.”*

2. The existence form and distribution of 3-AP molecules is not clear.

a) The authors argued that 3-AP additives play both role in anion fixation and suppressing the migration of anion vacancies on the crystal surface or at the grain boundaries. Please provide proof of the spatial distribution of 3-AP molecules in the perovskite film, such as Nano-IR or TOF-SIMS.

b) The amidino-based spacers have been widely studied for 2D Ruddlesden–Popper (RP) perovskites (Adv. Energy Mater. 2020, 2000694.; ACS Energy Lett. 2021, 6, 2072-2080). The authors have proved that 3AP molecules were arranged in parallel and anti-symmetrically between Pb-I frameworks. Hence, could 3-AP additives form the low-dimensional phase in this work?

c) The detailed preparation method and characterization of (3AP)PbI₄ single crystals should be included in the manuscript or supporting information.

Response: We have given high importance to this comment because the existence form and distribution of 3AP molecules in perovskite is essential to explore its interaction and mechanism.

(1) TOF-SIMS was performed to probe the profiles of the ionic species throughout the FAPbI₃ with 3API film on the TiO₂/FTO substrate as shown in **Figure R5**. The change trend of the signal representing 3AP⁺ is similar to FA⁺, I⁺ and Pb⁺ and decreases with their decreasing signal. Therefore, it can be speculated that 3AP is uniformly distributed in the bulk phase, which enables 3API to improve both grain boundaries and crystal surfaces during the growth of perovskite grains, thus improving the crystalline quality of perovskite films. Enlarged grains and vertical penetration of grain growth can be clearly observed in the cross-sectional SEM images of the device (further elaborated in the later section), which is consistent with the above speculation.

Figure R5 The characterization of TOF-SIMS for FAPbI₃ perovskite film with 3API.

We have updated the manuscript by including additional description and Figures (**Supplementary Fig. 8**) as following (See **Page 7**):

*“In addition, the Time-of-Flight Secondary Ion Mass Spectrometry (TOF-SIMS) was performed, which indicates a uniform distribution of 3AP⁺ throughout the film as shown in **Supplementary Fig. 8**.”*

(2) For the comment of “could 3-AP additives form the low-dimensional phase in this work”, we have done series of experiments for 3AP-doped films and single-crystal analysis on (3AP)PbI₄ perovskitoid. Our previous work indicate a unique crystal packing with corner-sharing and edge-sharing octahedra of inorganic frameworks and 3AP cations lying between adjacent inorganic layers in a parallel and antisymmetric manner. (**Chem. Mater.** **2022**, **34**, **1699-1709**) This implies that 3AP is impossible to form perovskite but adopt perovskitoid structure. So 2D perovskite structure can not be obtained in 3AP-doped FAPbI₃ films due to lattice mismatch between perovskite and perovskitoid structure. Indeed, we did not observe any the low-dimensional phase in 3AP-doped FAPbI₃ films from GIWAXS (**Figure 3a**), UV-Vis (**Figure 3d**), and ultrafast transient absorption spectroscopy (**Figure R6**).

Figure R6. The ultrafast transient absorption spectra of FAPbI₃ films with/without 3API.

(3) The detailed preparation method and characterization of (3AP)PbI₄ single crystals have been reported in our previous work (**Chem. Mater.** 2022, 34, 1699-1709). We also updated the manuscript as following:

“Synthesis of (3AP)PbI₄ single crystals. The synthesis and characterization of (3AP)PbI₄ single crystals can be found in previous work.¹” (Page 2 in the updated Supplementary information)

3. The stability test is insufficient to elucidate the marvelous stability.

a) For the operational stability test, in page 9 line 45, the word “approximately 30°C” is not scientific. Moreover, operational stability tests with the longer aging time under MPP tracking should be provided, such as 1000 h.

b) The authors refer to that 3-AP molecules realized anion-fixation and undercoordinated-Pb passivation in the form of hydrogen bonding. As we all know, the hydrogen bonding is much weaker than coordination and ionic bonding. Moreover, the 3-AP is a hydrophilic molecule. How can you achieve such a huge increase in ambient stability?

Response: We are thankful to the reviewers for the insightful comment. We have revised the title of the article to “*One-stone-for-two-birds strategy to attain >25% perovskite solar cells*”.

(1) For the “operational stability test”, we have corrected the word “approximately 30°C” to “28-34°C” (page 9 line 34, in the revised manuscript). For operational stability, unfortunately, we are unable to provide longer MPP tracking at this time because our laboratory equipment is not yet ready for assembly due to the impact of the COVID-19 pandemic prevents.

(2) For the comment of “how the hydrogen bonding strengthens device stability”, we have gone through previous reports. It has been well established that hydrogen bonding plays a very important role in the intrinsic stability perovskite [**Science**, 2019, 366, 1509, **Nature**, 2022, 603, 73, **Science**, 2022, 375, 71, **Nat. Commun.** 2022, 13, 3970, **Nat. Energy** 2019, 4, 408, **Nat. Chem.** 2015, 7, 703, **Adv. Mater.** 2019, 31, 1903721,

Angew. 2019, 58, 13912]. For example, during preparation of response to reviewers, Prof. Kai Zhu et al. used 3APy, which is quite similar to 3AP, to realize solar cell with PCE over 25% [**Nature**, 2022, DOI: 10.1038/s41586-022-05268-x]. They have confirmed the significant role of hydrogen bonding in the trap passivation and intrinsic stability of perovskite. In the early time, Chen et al. also found that hydrogen bonding between $[\text{PbI}_6]^{4-}$ and formamidinium-based ligands is beneficial for enhancing PCE and stability of the devices [**ACS Nano**. 2021, 15, 7811], as shown in **Figure R7**. Prof. Yang Yang also reported that hydrogen bonding of the amino hydrogen to surface iodide can optimize the carbonyl interaction with a lead antisite defect and improved the efficiency of a perovskite cell from 21 to 22.6% (**Science**, 2019, 366, 1509). All the above observations indicate that hydrogen bonding help better stabilize perovskite structure, thus leading to huge increase in ambient stability.

Figure R7 Aromatic formamidiniums enhances perovskite solar cells efficiency and improves device stability through hydrogen bonding interactions. Copyright 2021, American Chemical Society.

4. The authors illuminated that 3-AP molecules could form strong chemical bonds with the Pb-I framework. Since Pb and I determine the VB and CB of perovskite, 3-AP might influence the band alignment of perovskite film, which I think is one of the factors that enables the devices to achieve distinguished performance. The author should add the relevant experiments and discussions to the manuscript.

Response: Thanks to reviewer for the helpful comments. We have run the ultraviolet photo-electron spectroscopy (UPS) measurements for the films with/without 3API, as shown in **Figure R8**. The energy level alignment was given in **Figure R9**. The introduction of 3API reduced the VBM and CBM of perovskite from -5.46 eV and -3.92 eV to -5.55 eV and -4.01 eV, respectively. This indicates a better energy level alignment between the perovskite layer and the electron transport layer, which facilitates charge extraction and transfer, ultimately enabling the devices to achieve distinguished performance.

Figure R8 UPS measurements and band gaps of films with/without 3API.

Figure R9 The energy level alignment of perovskite solar cells.

Correspondingly, in the revised manuscript, we have added **Figure R8, 9** as **Supplementary Fig. 11, 12** and more explanation is added on **Page 8** in the revised manuscript as following:

“The ultraviolet photo-electron spectroscopy (UPS) test was employed to further illustrate the effect of 3API on the energy band alignment of perovskite (Supplementary Fig. 11 and 12). The introduction of 3API reduced the VBM and CBM of perovskite from -5.46 eV and -3.92 eV to -5.55 eV and -4.01 eV, respectively. The results indicate better energy level alignment between the perovskite layer and the electron transport layer, which is beneficial for charge extraction and transfer.”

5. In page 7 line 35, the authors refer to “a decrease in the surface roughness from 23.3 to 17.0 nm with 3API addition due to the stitched grain boundaries”, however, the SEM images reveal that there exist many cracks and pinholes at the grain boundaries of 3-AP treated film. Please provide a reasonable explanation.

Response: Thanks to the reviewer for pointing out the deficiencies in the SEM images. There are many cracks and pinholes in the initial SEM image, mainly due to the long irradiation of the film by the electron beam during the measurement. Therefore, we re-measured the SEM image of 3AP-doped film, as shown in **Figure R10**, and it was updated to the revised manuscript as **Figure 3b**.

Figure R10 SEM images of the 3AP-doped film.

6. The cross-sectional SEM images of the fabricated perovskite solar cells should be provided. And it is recommended to conduct further characterizations of the devices to confirm the effectiveness of 3-AP molecules, such as TPC, TPV testing.

Response: Thanks to reviewer for the comments. We fabricated complete perovskite solar cells and measured their cross-sectional SEM images (**Figure R11**). These cross-sectional images were updated to **Figure 3c** in the revised manuscript.

In addition, transient photocurrent (TPC) and transient photovoltage (TPV) were performed to confirm the effectiveness of 3AP⁺. Compared to the reference device, the device with 3API exhibited shorter charge extraction time (4.25 vs. 9.38 μs) and longer charge compounding time (11.2 vs. 3.07 μs). This indicates that the introduction of 3API can effectively improve the charge extraction process and suppress charge complexation (**Figure R12a and b**).

Figure R11 The cross-sectional SEM images of the PSCs.

Figure R12 (a) transient photocurrent (TPC) and (b) transient photovoltage (TPV) decay curves of PSCs for control and with 3API.

Correspondingly, we have included **Supplementary Fig. 13** and the relevant description in the revised manuscript as following (see **Page 8**):

*“Transient photocurrent (TPC) and transient photovoltage (TPV) were also performed to confirm the effectiveness of 3AP⁺ (**Supplementary Fig. 13a and b**). Compared to the reference device, the device with 3API exhibited a shorter charge extraction time (4.25 vs. 9.38 μs) and a longer charge compounding time (11.2 vs. 3.07 μs). This indicates that the introduction of 3API can effectively improve the charge extraction process and suppress charge complexation, which is consistent with the UPS results.”*

7. In the TRPL measurements, this expression of “was 5 times longer than that of the control (1.2 μs) is incorrect.

Response: Thanks for the reviewer’s comments. We have corrected “was 5 times longer than that of the control (1.2 μs)” to “was much longer than the control (1.2 μs)” on **page 7** in the revised manuscript.

Reviewer #2: This paper reports the positive effects of 3AP molecule addition on PV performance and stability for PSCs. Characterization data and DFT calculation results clearly support the positive effects of 3API. The result offers a chemical strategy for efficient stable PSCs so it is recommendable for publication in Nature Communication after clearing the following concerns.

Response: We appreciate the reviewer for taking the time to review our work. We also greatly thank the reviewer for the positive feedback on the work. We have added corresponding experimental and discussion are shown in **Figure R8, 9 and 13-16**. Meanwhile, we also improved the manuscript based the reviewer's insightful comments and listed point-to-point responses as shown below.

1. The main reason that authors insist on positive results is suppressing the formation of I⁻ vacancy by adding 3API. However, in Figure 3a, the 3AP also influences the crystal orientation of FAPbI₃ that FAPbI₃ with 3API shows a more oriented (001) direction than the control film. The orientation or crystallinity of film can influence the final device performance in addition to the trap density of the film. So additional experiments for post-surface passivation treatment are recommended to exclude orientation and crystallinity of neat FAPbI₃ film.

[<https://www.nature.com/articles/s41566-019-0398-2>

[<https://pubs.rsc.org/en/content/articlehtml/2021/xx/c9ee00751b>]

Response: We thank the reviewer for the kind suggestion. The introduction of 3API into FAPbI₃ perovskite, on the one hand, is to inhibit the formation of I⁻ vacancy defects, thus improving the performance and stability of the devices. On the other hand, we do not exclude the positive effect of 3API on the orientation and crystallinity of perovskite, as the reviewer stated. **Figure 3a** and **Supplementary Fig. 6** indicate that the 3API addition leads to stronger (100) orientation and crystallinity, which is one of the important reasons for the improved performance of the final devices. Therefore, we have revised the corresponding parts of the manuscript, as stated below:

“This provides strong evidence that 3AP can result in a complete conversion from undesired phases to perovskite 3C species, which is one of the key factors for the improvement of device performance.” (**Page 7** in the revised manuscript)

Of course, as for the post-surface passivation treatment, we simply dissolved 3API in IPA (2 mg/ml) and then spin-coated it on FAPbI₃ (PVK) films. The XRD patterns of the post-surface treated films did exhibit stronger (100) orientation and crystallinity (**Figure R13**). We further fabricated devices with structure FTO/c-TiO₂/PVK/3API/Spiro-OMeTAD/Au, and the device efficiency increased from 21.07% to 22.22% after 3API post-surface passivation treatment (**Figure R14**), indicating that 3API can still exert positive passivation effects on the perovskite films after post-surface passivation treatment.

Figure R13 The XRD patterns of films with/without 3API surface treatment.

Figure R14 The J-V curves of devices with/without 3API surface treatment.

2. The addition of 3API can change the work function of FAPbI₃ and result in an increase in V_{bi} . The UPS measurement for control film and FAPbI₃ film with 3API can support the result.

Response: Thanks for the reviewer's comments. The effect of introducing 3API on the work function and energy band alignment of FAPbI₃ perovskite is very important. We added UPS tests to reveal the effect of the introduction of 3API on the energy band alignment of the films (**Figure R8** and **R9**) and to present the change in the work function (**Figure R15**). The work function of containing 3API film increases from 3.97

to 4.13 eV, which supported the result that the addition of 3API can improve the V_{bi} of the devices.

Figure R15 The work function of control and with 3API sample.

3. In Figure 4b, spiro-OMeTAD normally shows poor thermal stability at 85 °C due to its low glass transition temperature. So it is hard to compare thermal stability between PSCs with spiro-OMeTAD. For estimation of thermal stability between control and target films, other HTMs should be used such as P3HT and PTAA instead of spiro-OMeTAD.

Response: We appreciate the kind comments of reviewer. Spiro-OMeTAD performed poorly in the thermal stability test, so we used PTAA as the HTL layer mentioned by the reviewer when testing the thermal stability. However, we neglected to note this in the manuscript, and we have added the corresponding statement as following:

“For the thermal stability of devices, 12 mg PTAA was dissolved in 1ml toluene, then 6 μ L Li-TFSI (340 mg Li-TFSI in 1 mL acetonitrile) and 6 μ L tBP were added. The PTAA solution was spin-coated on surface of the perovskite layer at 2000 rpm. for 30 s.” (Page 3 in the updated Supplementary information)

4. The detail for PL, TRPL (ex, excitation fluence...), M-S analysis (ex. Frequency...), and temperature-dependent conductivity measurements (ex, device structure....) should be described more in the experiments part.

Response: Thanks for reviewer’s suggestions. We have added the details for PL, TRPL, M-S analysis and temperature-dependent conductivity measurements to the experiments part. The details are described as follows:

In Material characterization part, we revised the statements of PL and TRPL as follows: *“Steady-state photoluminescence (PL) and time-resolved photoluminescence (TRPL) spectra were recorded with a PicoQuant FT-300 spectrometer. The excitation laser wavelength is 510 nm, the frequency is 40 MHz (for PL) and 0.5 MHz (for TRPL),*

and the detection wavelength range is 500-900 nm.” (Page 4 in the updated Supplementary information)

In the experimental section, the test frequency for M-S analysis was set to 50 kHz and the scan voltage range was 0-1.6V, which was revised on page 4 in the updated Supplementary information: “and the frequency was set to 200 kHz and the scan voltage range was 0-1.6V”.

As for the device structure for the temperature-dependent conductivity measurements was shown in Figure R16, and added as Supplementary Fig. 3 on page 5 in the revised manuscript as following: “(the schematic diagram of device structure is shown in Supplementary Fig. 3)”

Figure R16 The device structure for the temperature-dependent conductivity measurements.

REVIEWER COMMENTS

Reviewer #1 (Remarks to the Author):

Although most comments were answered, the concept of I- fixation and undercoordinated Pb²⁺ passivation by a single ligand in this work has been widely reported before (DOI:<https://doi.org/10.1039/D2EE02277J>; Adv. Energy Mater. 2022, 12, 2200537; Angew. Chem. 2022, 134, e202205012.). Moreover, in spite of the high efficiency and ambient stability achieved in this work, the mechanisms involved were not deeply studied, only superficially stated. From my point of view, innovation and mechanism investigation is of vital importance for the work to be published in Nature Communication, rather than just the high PCE performance. Therefore, considering that this work has not high guiding-significance, I still insist that this work is not appropriate for Nature Communication. I would like to make a few additional comments, as shown below.

1. The aging time of operational stability tests under MPP tracking is too short. This work focuses on the inhibition of I- ions migration by 3-AP ligand, which is more important for improving operation stability compared with ambient stability. However, the test time of operation stability in this work is only 200 hours.

2. In Page 9 line 299, the authors referred to that “Compared with the control cell, the 3AP-based cell exhibited a smaller FF loss during aging.” However, the manuscript did not present the corresponding photovoltaic parameters. Moreover, the authors did not thoroughly study on the mechanism of stability improvement.

3. We have carefully studied the articles published by Nature Communication in the past two years, and found that more articles of perovskite solar cells are in the research of mechanism, such as Nat. Commun. 2022, 13, 2868; Nat. Commun. 2021, 12, 2853.; Nat. Commun. 2021, 12, 6394; Nat. Commun. 2021, 12, 3383; Nat. Commun. 2021, 12, 7; Nat. Commun. 2022, 13, 4891; Nat. Commun. 2022, 13, 4417. However, regardless of the high PCE performance, this paper is a little superficial and can not give the readers deep inspiration.

Reviewer #2 (Remarks to the Author):

The revised manuscript has mostly addressed the concerns. However, in the discussion of long-term stability, initial PCEs should be represented. In particular, if the device used a different HTM from representing architecture in Figure 2a, the initial PCE including parameters (J_{sc} , V_{oc} , FF) should be represented to compare to the control device using Spiro-OMeTAD. Because the Spiro-OMeTAD is weak to estimate the long-term stability of perovskite solar cells, PV performance using the alternatives is meaningful in this field. If this part is supplemented, I think it is acceptable without additional evaluation.

List of point-to-point response of reviewers' comments

Reviewer #1: Although most comments were answered, the concept of I⁻ fixation and undercoordinated Pb²⁺ passivation by a single ligand in this work has been widely reported before (DOI:<https://doi.org/10.1039/D2EE02277J>; Adv. Energy Mater. 2022, 12, 2200537; Angew. Chem. 2022, 134, e202205012.). Moreover, in spite of the high efficiency and ambient stability achieved in this work, the mechanisms involved were not deeply studied, only superficially stated. From my point of view, innovation and mechanism investigation is of vital importance for the work to be published in Nature Communication, rather than just the high PCE performance. Therefore, considering that this work has not high guiding-significance, I still insist that this work is not appropriate for Nature Communication. I would like to make a few additional comments, as shown below.

Response: We appreciate the reviewer for taking the time to review our work. And we also thank reviewer for the insightful comments to help us improve the quality of the manuscript, for which we have given this comment significant thought and effort.

Firstly, we agree that the I⁻ fixation and undercoordinated Pb²⁺ passivation are not a novel concept. We consulted carefully the research papers from the past decade or so. The concept was reported as early as 2014 and has been well followed in the following years (ACS Nano 2014, 8, 9815; Nat. Commun. 2015, 6, 7081; Nat. Commun. 2015, 6, 10030; Nat. energy 2016, 1, 16149; Nat. energy 2017, 2, 17102; Adv. Mater. 2017, 1606774; Adv. Mater. 2018, 1706576; Energy Environ. Sci. 2018, 11, 3480). The widely research has confirmed. These articles, including the 3 articles mentioned above, more or less contribute to the fact that “I⁻ fixation and undercoordinated Pb²⁺ passivation” is very much in need for the developments of perovskite solar cells. As a result, passivation strategies are becoming more and more effective, as is our work.

Secondly, regarding the innovation of this work, I believe quite few people have seen any reports of individual ligand molecules lying flat in a lead-iodine inorganic framework with a perovskite structure, let along how this packing style affects perovskite stacking and the following charge transport behavior. So, developing such ligand molecules and unlocking the underlying mechanism for highly-efficient solar cells are undoubtedly innovative. In fact, up to now, almost all single ligands that have been used to modulate perovskite solar cells have been reported in the form of vertical or tilted or unclear specific distribution structures. While this packing style of 3AP lying flat between the inorganic framework is the first report in the field of perovskite solar cells, and our results have shown superior ability in anion fixation and suppressing the migration of anion vacancies in contrast to previous results (Figure 1), and therefore providing valuable knowledge and advances for minimizing anion-vacancy defects in PSCs to push PCE more close to 26%. Therefore, we think our work is full of innovation and it is worth to be published in Nature Communication.

Thirdly, based on the above new arrangement of amidino-based molecules, we think the reviewer may not fully understand the changes and effects of this particular

arrangement, which is definitely different from the previously reported conventional organic ligands. The 3AP molecules were arranged in parallel and anti-symmetrically between Pb–I frameworks with a short interlayer distance of 3.45 Å, which is much shorter than that obtained for previously reported ligands, leading to a unique coordination within the crystals (**Table S2**). And the theoretical calculations (**Figure 1b,c, Figure R13,14**) and experiments (**Figure 1f, Figure S6**) also demonstrate that 3AP has a stronger synergistic effect than previously reported vertical or tilted ligands.

Fourthly, regarding the mechanism study, we also added tests such as in situ PL of the film crystallization process and photocurrent of the complete device in the operating state, which will be analyzed and discussed later. Therefore, we propose the mechanisms of in situ anchoring of iodide ions and in situ defect passivation. Of course, the improvement mechanism of 3AP on device performance is not only limited to in situ anchoring and passivation.

Fifthly, regarding the guiding significance of this work, we innovatively proposed a class of amidino-based molecules with a special arrangement structure, breaking the perception of conventional organic ligand molecules. Moreover, we propose the crystallization regulation mechanism of in situ anchoring and passivation, which emphasizes the importance of controlling the in situ crystallization process of thin films. Overall, this work will provide extremely important references in many fields such as the search for new and more efficient organic ligands, the design of perovskite crystal structures, FA-based crystallization regulation, in situ characterization techniques, and the photocurrent properties of devices in the operating state. Due to the wide application area of the results of this work, it should be of interest to the broad readership of the Nature Communications.

Based on the above mentioned, we revisited the important role of 3AP as a novel amidino organic ligand for perovskite performance enhancement. Although I-fixation and uncoordinated Pb²⁺ passivation have been mentioned in recently published articles, we have also carefully compared these literatures. Peng et al. introduced the new amine salt APC to the surface and superficial layers of perovskite, thus improving the film quality and passivation defects (DOI: <https://doi.org/10.1039/D2EE02277J>). However, APC is simply introduced to the surface and shallow layers of perovskite as part of the antisolvent. In addition, a more in-depth **passivation mechanism is not given in the article** but only a general proposed interaction schematic is presented (**Figure R1a**). Ma et al. introduced the natural molecule indigo into the antisolvent as well, utilizing the carbonyl and amino groups in the indigo molecule to interact with the Pb²⁺ and I⁻ ions through Lewis acid base interaction (**Adv. Energy Mater.** 2022, 12, 2200537) (**Figure R1b**). Our previous work (**Angew. Chem.** 2022, 134, e202205012), Che et al. employed hydrazide organic molecules (benzoyl hydrazine (BH), formohydrazide (FH) and benzamide (BA)) to passivate all-inorganic CsPbI₃ perovskite, emphasizing the positive role of hydrazide derivatives in all-inorganic chalcogenides (**Figure R1c**).

In our work, we report a new class of amidino-based organic molecules, 3APX (X = I, Br, Cl), and their introduction into perovskite precursors to achieve full spatial

defect passivation during thin film preparation. Crystal structure analysis revealed a special "parallel-antisymmetric" arrangement of 3AP with the inorganic $[\text{PbI}_6]^{4-}$ framework (**Figure R1d**), **which is not observed in most of the organic ligands reported so far**. With the special conformation of the amidino group and pyridine, 3AP has stronger interactions than the common amine salt molecules, including but not limited to surface adsorption energy, defect formation energy, I^- ion migration energy barrier, ion migration activation energy etc., as mentioned in our manuscript (**Figure 1**). We also found that the introduction of 3AP not only enables I^- fixation and undercoordinated Pb^{2+} passivation of the final film, but also in situ modulation of the film formation process, resulting in locking of the halogen anion and in situ passivation of stacking defects during crystallization, as will be specified in the following discussion.

Figure R1 (a) Schematic illustration of the proposed interaction mechanism between APC and the perovskite (Copyright from DOI: 10.1039/D2EE02277J). (b) Schematic illustration of the passivating mechanism between Indigo and mixed hybrid perovskite (Copyright from Adv. Energy Mater. 2022, 12, 2200537). (c) The device architecture and presumed interaction models of BH, FH and BA towards perovskite films (Copyright from Angew. Chem. 2022, 134, e202205012). (d) The structure of (3AP)PbI₄ (Figure S1 in the Supplementary Information).

In addition, we are also concerned about the submission and publication dates of these articles mentioned above, which are similar to or even later than our submission date (**Figure R2**). We submitted the manuscript to Nature communications on 12 May, 2022 (**Figure R2a**), and the preprint version went live on Nature portfolio on 20 May. To date, the preprinted manuscript has received **547 views and 100 downloads** (**Figure R3**), which exhibits the attractiveness and reference potential of our work to a wide readership in the relevant field. (<https://www.researchsquare.com/article/rs-1648855/v1>).

Figure R2 (a) Submission history and timeline for this work in Nature Communications. (b-d) Submission history and timeline for articles (DOI: <https://doi.org/10.1039/D2EE02277J>; Adv. Energy Mater. 2022, 12, 2200537; Angew. Chem. 2022, 134, e202205012.).

Figure R3 Preprint version of the manuscript on natureportfolio and the number of views and downloads.

Sincerely, we are encouraged by the reviewer's overall assessment of our manuscript to improve our paper. We added relevant experiments and characterizations including in situ photoluminescence (PL) spectroscopy (**Figure R4**), photocurrent mapping (**Figure R5, R6, R7**) and KPFM maps (**Figure R9**) to further understand the synergistic mechanism of 3AP with perovskite. We sincerely invite reconsider our work for publishing in *Nature Communications*.

Figure R4 shows the in situ PL spectra of the films during spinning and thermal annealing. During spinning, both films exhibit negligible emission (**Figure R4a, b**,

left). During the annealing process (**Figure R4a, b, right**), the suggesting no illuminating α -FAPbI₃ phase shows a rapid formation as evidenced from a quick rise of PL intensity. The following sharp decrease of PL intensity can be attributed to solvent volatilization, film crystallization restructuring, and negative temperature coefficient. (**Sci. Adv.** 2021, 7, eabj1799; **Phys. Chem. Chem. Phys.** 2014, 16, 22476; **Nat. Mater.** 2014, 13, 476) Note that the control film exhibits a continuous but a slow decrease of PL intensity with further thermal annealing (**Figure R4c**), which is indicative of the formation of stacking defects. The 3API addition not only retarded the perovskite crystallization process but also inhibited the formation of stacking defects, as confirmed by the slowly increased PL intensity with further thermal annealing (**Figure R4d**). This indicates that in situ crystallization modulation and defect passivation can be realized through the strong anchoring effect of 3API with [PbI₆]⁴⁻ during the formation of the films.

Figure R4 (a, b) The color mapping of the captured PL spectra during spinning (left) and thermal annealing (right) for the control and with 3AP films, respectively. **(c, d)** The PL intensity extract from **a** and **b**.

The PL intensity and photocurrent mappings on the working PSCs were recorded by using a laser-scanned and time-resolved PL microscopy coupled with a photocurrent detection module as we reported previously.⁴⁹ We collected the PL intensity and local photocurrent mapping within a $10.5 \times 10.5 \mu\text{m}$ area for both devices (**Figure R5**). **Figure R6a, b** shows the PL intensity mapping, where the distribution of grain size and shape is clearly identifiable. The devices with 3API show a larger perovskite grain size (**Figure R7a**) and are coupled with a stronger PL intensity than the control devices, which is consistent with the SEM (**Figure 3b**), PL (**Figure 3d**), TRPL (**Figure 3e**) and PL mapping (**Supplementary Figure 19**) results. Furthermore, we observed fewer photons from larger grains for both devices, which indicates more efficient charge separation and extraction. The photocurrent mapping on the same area are shown in Fig. 4e, f, right. The devices with 3API exhibit higher photocurrents compared to the control (**Figure R7b**), which further indicates less stacking defects

with 3API. We also performed statistical histograms for the different photocurrent values in the photocurrent images. It is clear that the percentage of photocurrent values >16 nA in the devices increase from 43% to 55% after the introduction of 3API (**Figure R6e, f**). Both in situ observation during film formation and ex situ analysis on the working devices explain well the importance of in situ passivation of stacking defects during the formation for highly efficient charge separation and extraction in the working devices.

Figure R5 Schematic presentation of the laser-scanned and time-resolved PL microscopy coupled with a photocurrent detection module.

Figure R6 PL intensity and Photocurrent of the PSCs with or without 3AP. (a, b) PL intensity images. (c, d) Photocurrent images collected on the same area as in (a) and (b). (e, f) Statistical histograms for the different photocurrent values extracted from the photocurrent images.

Figure R7 Statistical Analysis. (a) Histogram of grain size statistics extracted from PL intensity images. (b) Scatter plot of photocurrent statistics extracted from photocurrent images.

Based on the above discussed, we added **Figure R4** and **R6** to the manuscript as **Figure 4 (Figure R8)**, and added **Figure R5** and **R7** to Supplementary Information as **Figure S25, S26**, respectively. We also added the corresponding discussion in the manuscript as follows:

“Stacking Defects

In sought to better understand how the 3AP addition influences stacking defects of perovskites, in situ PL spectroscopy and photocurrent mapping were performed during perovskite fabrication process and for devices when working under

illumination, respectively. **Fig. 4a, b** shows the *in situ* PL spectra of the films during spinning and thermal annealing. During spinning, both films exhibit negligible emission (**Fig. 4a, b, left**). During the annealing process (**Fig. 4a, b, right**), the suggesting no illuminating α -FAPbI₃ phase shows a rapid formation as evidenced from a quick rise of PL intensity. The following sharp decrease of PL intensity can be attributed to solvent volatilization, film crystallization restructuring, and negative temperature coefficient.^{46–48} Note that the control film exhibits a continuous but a slow decrease of PL intensity with further thermal annealing (**Fig. 4c**), which is indicative of the formation of stacking defects. The 3API addition not only retarded the perovskite crystallization process but also inhibited the formation of stacking defects, as confirmed by the slowly increased PL intensity with further thermal annealing (**Fig. 4d**). This indicates that *in situ* crystallization modulation and defect passivation can be realized through the strong anchoring effect of 3API with [PbI₆]⁴⁻ during the formation of the films.

The PL intensity and photocurrent mappings on the working PSCs were recorded by using a laser-scanned and time-resolved PL microscopy coupled with a photocurrent detection module as we reported previously.⁴⁹ We collected the PL intensity and local photocurrent mapping within a 10.5 × 10.5 μm area for both devices (**Supplementary Fig. 25**). **Fig. 4e, f, left** shows the PL intensity mapping, where the distribution of grain size and shape is clearly identifiable. The devices with 3API show a larger perovskite grain size (**Supplementary Fig. 26a**) and are coupled with a stronger PL intensity than the control devices, which is consistent with the SEM (**Fig. 3b**), PL (**Fig. 3d**), TRPL (**Fig. 3e**) and PL mapping (**Supplementary Fig. 19**) results. Furthermore, we observed fewer photons from larger grains for both devices, which indicates more efficient charge separation and extraction. The photocurrent mapping on the same area are shown in **Fig. 4e, f, right**. The devices with 3API exhibit higher photocurrents compared to the control (**Supplementary Fig. 26b**), which further indicates less stacking defects with 3API. We also performed statistical histograms for the different photocurrent values in the photocurrent images. It is clear that the percentage of photocurrent values >16 nA in the devices increase from 43% to 55% after the introduction of 3API (**Fig. 4g, h**). Both *in situ* observation during film formation and *ex situ* analysis on the working devices explain well the importance of *in situ* passivation of stacking defects during the formation for highly efficient charge separation and extraction in the working devices.” (Page 9, 10, in the revised manuscript)

In view of this, we have also emphasized the importance of *in situ* modulation in the abstract and conclusion section, as follows:

“which both *in situ* anion-fixation and associated undercoordinated-Pb passivation are *in situ* achieved during crystallization” (Page 1, in the revised manuscript)

“*in situ* PL” (Page 13, in the revised manuscript)

Figure R8 Stacking defects during film formation and in the working devices. a and b, In situ PL spectra of films with or without 3API during spin coating and annealing. **c and d,** The PL intensity extract from **a** and **b**. **e and f,** PL intensity and photocurrents of the working devices with or without 3API. **g and h,** The histogram of photocurrent statistics. (**Figure 4**, in revised manuscript)

Kelvin probe force microscopy (KPFM) analysis indicates a higher electronic chemical potential of perovskite with the 3AP addition (**Figure R9**). The higher electronic chemical potential and reduced energy level (**Figure S13, 14**) indicate a less n-type surface, which can facilitate the hole extraction in devices [**Science**, 2019, 366, 1509].

Figure R9 KPFM images of perovskite films with and without 3API.

Figure R9 was added to the manuscript as **Figure S16**, and we have also added the corresponding discussion in the revised manuscript as follows:

“Kelvin probe force microscopy (KPFM) analysis indicates a higher electronic chemical potential of perovskite with the 3AP addition (Supplementary Fig. 16). The higher electronic chemical potential and reduced energy level (Supplementary Fig. 13 and 14) indicate a less n-type surface, which can facilitate the hole extraction in devices.¹⁹” (Page 8, 9, in revised manuscript)

Furthermore, we have added measurements of MPP tracking based on the suggestions from the reviewer. The detailed responses to each point and the related modifications in the revised manuscript are shown below.

1. The aging time of operational stability tests under MPP tracking is too short. This work focuses on the inhibition of I⁻ ions migration by 3-AP ligand, which is more important for improving operation stability compared with ambient stability. However, the test time of operation stability in this work is only 200 hours.

Response: Thanks to the reviewers' comments, we have redone the operation stability tests and the updated MPP tracking is given in **Figure R10**. The 3AP-based solar cells exhibited a 5% decrease in their initial efficiency after 510 h of continuous operation. Meanwhile, the PCE of the control device decreased by 16% after only 240 h.

Figure R10 The operation stability of the control and 3AP-based devices.

We updated **Figure 5 (Figure R11)** and corresponding descriptions are as follows:

“Finally, we checked the operational stability of encapsulated solar cells by performing maximum power point (MPP) tracking in the ambient (25 °C and 30–40%

relative humidity) following the ISOS-L-1 stability protocol.⁵⁰ The 3AP-based solar cells exhibits a 5% decrease of their initial efficiency after 510 h. While the PCE of the control device decreased by 16% after only 240 h (Fig. 5d).” (Page 10,12, in the revised manuscript)

“the device with encapsulation retained 95% of its initial efficiency after > 500 h of operation at the maximum power point under continuous light irradiation in ambient.” (Page 1, in the revised manuscript)

Figure R11 Stability studies of perovskite films and solar cells. a, XRD patterns of the control and 3AP-based perovskite films recorded before and after aging for 74 days. **b**, PCE evolution observed for the control and 3AP-based devices stored without encapsulation at 85 °C under a nitrogen atmosphere. **c**, PCE evolution observed for the control and 3AP-based devices stored without encapsulation at ambient conditions (25 °C and 30–40% relative humidity). **d**, Stabilized MPP values recorded for the control and 3AP-based devices with encapsulation under continuous light irradiation with a white LED lamp at 100 mW cm⁻² in ambient (25 °C and 30–40% relative humidity) (Figure 5, in revised manuscript)

2. In Page 9 line 299, the authors referred to that “Compared with the control cell, the 3AP-based cell exhibited a smaller FF loss during aging.” However, the manuscript did not present the corresponding photovoltaic parameters. Moreover, the authors did not thoroughly study on the mechanism of stability improvement.

Response: Thanks to the reviewers' comments. We present the $J-V$ curves of the devices before and after aging (Figure R12a) and the corresponding optoelectronic parameters (Table R1). The FF of the reference devices decays by 21.2% after aging, while the device with the addition of 3AP decays by only 6.6%. It can also be seen from Figure R12b that the 3AP-based device exhibits a smaller FF loss, while V_{oc} and J_{sc} show a slight attenuation (Figure R12c, d).

Figure R12. (a) The $J-V$ curves of the devices before and after aging. (b-d) Long-term stability of photovoltaic parameters.

Table R1 The photovoltaic parameters of the devices before and after aging.

		PCE (%)	V_{oc} (V)	J_{sc} (mA cm^{-2})	FF (%)
Control	Initial	22.11	1.096	25.24	79.91
	Aging	17.37	1.097	25.14	62.96
With 3API	Initial	24.02	1.158	25.32	81.86
	Aging	22.31	1.155	25.25	76.45

We added **Figure R12a** and **Table R1** to the Supplementary information as **Figure S28 and Table S6**, and revised the corresponding part of the manuscript.

“Compared with the control cell, the 3AP-based cell exhibited a smaller FF loss during aging (Supplementary Fig. 28 and Table 6).” (Page 10, in the revised manuscript)

For the mechanism of stability improvement, it mainly comes from the hydrogen bonding and in situ anchoring of 3AP with perovskite. Regarding hydrogen bonding, NMR (**Figure 1d**) and FTIR (**Figure S4**) in our manuscript have demonstrated the formation of strong hydrogen bonds between 3AP with the Pb-I inorganic framework and its stabilizing effect on I^- anions at the crystal surface and grain boundaries. Therefore, hydrogen bonding is one of the important mechanisms to improve the stability of the device. It has been well established that hydrogen bonding plays a very important role in the intrinsic stability perovskite [*Science*, 2019, 366, 1509, *Nature*, 2022, 603, 73, *Science*, 2022, 375, 71, *Nat. Commun.* 2022, 13, 3970, *Nat. Energy* 2019, 4, 408, *Nat. Chem.* 2015, 7, 703, *Adv. Mater.* 2019, 31, 1903721, *Angew.* 2019, 58, 13912]. As for in situ anchoring, in situ PL (**Figure R4**) demonstrate in situ anchoring of anions and in situ passivation of defects by 3AP during film crystallization. The obtained high quality and low defect films are another key mechanism for device stability improvement.

Furthermore, we added DFT theoretical calculations to account for the strong synergistic effect of 3AP with perovskite. **Figure R13** shows the crystal structure model constructed by the first-principles calculations in the frame of density functional theory (DFT) with the program package CASTEP. The calculations indicate that 3AP has the highest adsorption energies on the perovskite surface, formation energies of I^- vacancies in the perovskite lattice, and migration energy barrier of I^- diffusion on FAI-terminal FAPbI_3 (001) surface and MAI-terminal MAPbI_3 (001) surface (**Figure 1c**, **Table S3** and **Figure R14**). These results indicate that the strong 3AP–perovskite coordination plays the role in anion fixation and suppressing the migration of anion vacancies.

We also added **Figure R13** and **R14** to the Supplementary information as **Figure S2 and S3**, and revised the corresponding part of the manuscript as follows:

“(Fig. 1b and Supplementary Fig. 2)” and *“(Supplementary Fig. 3 and Table 3)”* (Page 3, in the revised manuscript)

Figure R13. The crystal structure model constructed by the first-principles calculations in the frame of density functional theory (DFT) with the program package CASTEP. (a) Legend and crystal coordinates. (b) MAPbI₃-001. (c) FAPbI₃-001. (d-h) 3AMP, 3AMPY, 3AP, BA and PEA interact with FAPbI₃ terminals, respectively.

Figure R14. The energy barrier of I⁻ diffusion on FAI-terminal FAPbI₃ (001) surface and MAI-terminal MAPbI₃ (001) surface.

As well as theoretical calculations, we have experimentally verified the effective inhibition of ion migration by 3AP in films and devices. The activation energy (E_a) of ion migration was experimentally evaluated by measuring the temperature-dependent conductivity of perovskite films. The highest E_a value of 0.198 eV was demonstrated for the film with 3AP (**Figure S6**, **Figure 1f**). Cross-sectional SEM images and

corresponding Energy Dispersive Spectroscopy (EDS) of the devices were performed to monitor the ion migration of each functional layer before and after devices aging (**Figure R15**). The interface between the layers in the initial device can be clearly observed from **Figure R15a, b**. However, after aging, the control device exhibits a more dispersed interface than the optimized device, especially a more pronounced migration of the I-element toward Spiro-OMeTAD layer (**Figure R15c, d**). This indicates that 3AP can firmly lock I⁻ ions and inhibit ion migration during the long-term aging of the devices.

The above results again demonstrate the strong hydrogen bonding and strong in situ anchoring effect of 3AP on perovskite. These are the critical elements for device stability improvement.

Figure R15 The cross-sectional SEM images and corresponding Energy Dispersive Spectroscopy (EDS) of the devices with or without 3AP before and after aging.

We also added **Figure R15** to the Supplementary information as **Figure S27**. and revised the corresponding part of the manuscript as follows:

*“The cross-sectional SEM images and corresponding Energy Dispersive Spectroscopy (EDS) of complete cells before and after aging indicate less I⁻ migration toward Spiro-OMeTAD layer (**Supplementary Fig. 27**) with the 3AP addition, which is due to strong hydrogen bonding and anchoring effect of 3AP on perovskite.”* (Page 10, in the revised manuscript)

3. We have carefully studied the articles published by Nature Communication in the past two years, and found that more articles of perovskite solar cells are in the research of mechanism, such as Nat. Commun. 2022, 13, 2868; Nat. Commun. 2021, 12, 2853.; Nat. Commun. 2021, 12, 6394; Nat. Commun. 2021, 12, 3383; Nat. Commun. 2021, 12, 7; Nat. Commun. 2022, 13, 4891; Nat. Commun. 2022, 13, 4417. However, regardless of the high PCE performance, this paper is a little superficial and can not give the readers deep inspiration.

Response: We are thankful to the reviewers for the insightful comment and we have carefully studied the mentioned above Nature Communications.

Cacovich et al. investigated the interface recombination dynamics related to the passivation approach in perovskite solar cells based on the introduction of large cations such as Cl-PEAI and F-PEAI by using multidimensional photoluminescence imaging. (Nat. Commun. 2022, 13, 2868). Haque et al. elucidate the degradation mechanism of 2D/3D tin perovskite films based on $(\text{PEA})_{0.2}(\text{FA})_{0.8}\text{SnI}_3$ and establishing a cyclic degradation mechanism: $\text{ASnI}_3 \rightarrow \text{SnI}_4 \rightarrow \text{HI} \rightarrow \text{I}_2 \rightarrow \text{ASnI}_3 \rightarrow \text{SnI}_4$. They employed XRD, UV-Vis, NMR and DFT characterization to support the degradation mechanism and highlight the important role of SnI_4 . (Nat. Commun. 2021, 12, 2853). Dai et al. studied the energy barrier of 2D perovskite formation from ortho-, meta- and para-isomers of (phenylene)di(ethylammonium)iodide (PDEAI_2) that were designed for tailored defect passivation. They showed that the most sterically hindered ortho-isomer can prevent the formation of in-plane 2D perovskite and passivate the deep and shallow energy levels through temperature-dependent PL images, GIWAXS and theoretical calculations. (Nat. Commun. 2021, 12, 6394). Grätzel et al. reported a multimodal host-guest complexation strategy to overcome this challenge using a crown ether, dibenzo-21-crown-7, which acts as a vehicle that assembles at the interface and delivers Cs^+ ions into the interior while modulating the material. Ultimately, the surface and bulk phase defects are passivated and the perovskite is stabilized by a synergistic effect of the host, guest, and host-guest complex (Nat. Commun. 2021, 12, 3383). Huang et al. reported metallic surface doping of perovskite by some metal ions such as silver, strontium, cerium ions, and illustrated the effect of such n-doping on the carrier concentration by experiments and calculations. They also explain the difficulty for the bulk doping of perovskites, because most metal ions cannot incorporate into the crystal structure of metal halide perovskite. Finally, they propose that the nonuniform distribution of metal ion dopants in perovskite polycrystalline films may strongly impact halide perovskite optoelectronics in many different ways (Nat. Commun. 2021, 12, 7). Wei et al. investigated the effect of moisture during perovskite crystallization by established air-exposure-free techniques. They find the moisture treatment can accelerate the quasi-solid-solid reaction between organic salts and PbI_2 , and enables a spatially homogeneous intermediate phase and translates to high-quality perovskites with much-suppressed defects. (Nat. Commun. 2022, 13, 4891) Pang et al. revealed that the degradation of formamidinium-containing perovskites in aliphatic amines environment results from the transamination reaction of formamidinium cation and aliphatic amines along with the formation of ammonia. They proposed the strategy of ammonia post-healing treatment and finally prepared highly uniform and compact FA-based perovskite films. (Nat. Commun. 2022, 13, 4417). In summary, in addition to the mechanism, these articles also focus equally on the performance of the perovskite cells. To this end, we also add more characterizations to support the synergistic effect of 3AP (**Figure R4, R5, R6, R7** and **Figure R13, R14, R15**) and propose mechanisms for in situ I^- fixation and defect modulation.

More importantly, in this work, we report for the first time a completely new crystal

structure based on a special arrangement of 3AP between the lead-iodine inorganic framework (parallel-antisymmetric), and we find that the introduced 3AP is present in the lead-iodine inorganic framework in a flat lying manner, which is quite different from all the organic ligands reported so far that are mainly in a vertical or tilted manner (**Figure S1**). Secondly, through calculations and experiments we found that this particular arrangement of 3AP gives it a stronger synergistic effect with perovskite compared to conventional organic ligands. This allows 3AP to have one of the best iodine fixation and defect passivation effects (**Figure 1**). Thirdly, we observe for the first time the in situ defect stacking of perovskite from the dynamics and propose a mechanism for the in situ realization of I⁻ fixation and defect modulation by 3AP. In addition, PL intensity variations and photocurrent enhancement phenomena were observed in the operating devices (**Figure 4**).

Based on the above findings and mechanisms, we have obtained high quality perovskite films and high efficiency perovskite devices. Our innovative proposal of a new class of amidino-based organic molecules, which breaks the limitation of the conventional use of amine-based organic molecules to modulate perovskite. It is worth noting that the modulation of in situ dynamics crystallization of perovskite will be a very important factor, and we firmly believe that this work will provide a strong reference for the improvement and development of perovskite device performance.

Reviewer #2: The revised manuscript has mostly addressed the concerns. However, in the discussion of long-term stability, initial PCEs should be represented. In particular, if the device used a different HTM from representing architecture in Figure 2a, the initial PCE including parameters (J_{sc} , V_{oc} , FF) should be represented to compare to the control device using Spiro-OMeTAD. Because the Spiro-OMeTAD is weak to estimate the long-term stability of perovskite solar cells, PV performance using the alternatives is meaningful in this field. If this part is supplemented, I think it is acceptable without additional evaluation.

Response: We appreciate the reviewer for taking the time to review our work. The initial efficiency parameters regarding the long-term stability of the device are shown in **Figure R16** (**Figure R12a**) and **Table R1**.

Figure R16 Initial and aging J - V curves for devices with long-term stability testing.

Table R1 The photovoltaic parameters of the devices before and after aging.

		PCE (%)	V_{oc} (V)	J_{sc} (mA cm ⁻²)	FF (%)
Control	Initial	22.11	1.096	25.24	79.91
	Aging	17.37	1.097	25.14	62.96
With 3API	Initial	24.02	1.158	25.32	81.86
	Aging	22.31	1.155	25.25	76.45

In addition, the initial efficiencies of the devices using PTAA instead of Spiro-OMeTAD as HTL in the thermal stability tests were 21.59% and 19.45% with and without 3API, respectively, and the corresponding parameters have been added to the Supplementary Information as follows:

“(with an initial PCE of 21.59%, a V_{oc} of 1.104 V, a J_{sc} of 25.43 mA/cm², a FF of 76.89% for With 3API, and an initial PCE of 19.45%, a V_{oc} of 1.063 V, a J_{sc} of 25.21 mA/cm², a FF of 72.62% for Control.)” (Page 3, in revised Supplementary Information)

REVIEWERS' COMMENTS

Reviewer #1 (Remarks to the Author):

On the one hand, in this reply, the authors added more substantial and convincing experiments and characterizations including in situ photoluminescence (PL) spectroscopy, photocurrent mapping and KPFM maps, which can more intuitively illuminate the synergistic mechanism of 3AP with perovskite. On the other hand, the authors carried out again the operation stability test and prolonged the test time to 500 h, which is more convincing. In addition, the authors summarized and emphasized the innovation through sufficient literature research. The point of view in which the AP molecules interact with the [PbI₆]⁴⁻ octahedron in a special "parallel-antisymmetric" arrangement to achieve I⁻ fixation and undercoordinated Pb²⁺ passivation is quite insightful. Overall, I think that this revision is much more profound than before. Therefore, I recommend this paper to be published in Nature Communication.

Reviewer #2 (Remarks to the Author):

The 2nd revised manuscript offers the required information for stability tests. But I recommend mentioning the use of PTAA in the main text for readers to avoid misunderstanding. If this part will be reflected, my review would not be required further.

List of point-to-point response of reviewers' comments

Reviewer #1 (Remarks to the Author):

On the one hand, in this reply, the authors added more substantial and convincing experiments and characterizations including in situ photoluminescence (PL) spectroscopy, photocurrent mapping and KPFM maps, which can more intuitively illuminate the synergistic mechanism of 3AP with perovskite. On the other hand, the authors carried out again the operation stability test and prolonged the test time to 500 h, which is more convincing. In addition, the authors summarized and emphasized the innovation through sufficient literature research. The point of view in which the AP molecules interact with the [PbI₆]⁴⁻ octahedron in a special "parallel-antisymmetric" arrangement to achieve I⁻ fixation and undercoordinated Pb²⁺ passivation is quite insightful. Overall, I think that this revision is much more profound than before. Therefore, I recommend this paper to be published in Nature Communication.

Response: We appreciate the reviewers' comments on this work, which are very important to improve the quality of the manuscript. It is also a great encouragement to our work. Once again, we thank the reviewers for their valuable comments.

Reviewer #2 (Remarks to the Author):

The 2nd revised manuscript offers the required information for stability tests. But I recommend mentioning the use of PTAA in the main text for readers to avoid misunderstanding. If this part will be reflected, my review would not be required further.

Response: We thank the reviewer for the thoughtful suggestions and we have mentioned the use of PTAA in the revised manuscript as follows:

“We used Poly[bis(4-phenyl)(2,4,6-trimethylphenyl)amine] (PTAA) instead of Spiro-OMeTAD and the organic passivation layer was removed to avoid introducing sources of instability.” (Page 7, in the revised manuscript)